# Auto-Arena: Automating LLM Evaluations with Agent Peer Battles and Committee Discussions

## Abstract

As LLMs continuously evolve, there is an urgent need for a reliable evaluation method that delivers trustworthy results promptly. Currently, static benchmarks suffer from inflexibility and unreliability, leading users to prefer human voting platforms like Chatbot Arena. However, human evaluations require significant manual effort. To address this, we propose the `Auto-Arena`, an innovative framework that automates the entire evaluation process using LLM-powered agents. Firstly, an LLM examiner generates questions. Then, two LLM candidates engage in a multi-round peer battle based on individual questions, aiming at revealing their true performance differences. Finally, a committee of LLM judges collaboratively discusses and decides the winner, reducing bias and enhancing fairness. During the peer battles, we observe intriguing scenarios where the LLM candidates display competitive behaviors and even learn from the opponents. In our extensive experiments involving 15 recent LLMs, `Auto-Arena` shows a 92.14% correlation with human preferences, surpassing all previous expert-annotated benchmarks without any manual efforts. As a result, `Auto-Arena` offers a promising alternative to current human evaluation platforms for evaluating LLMs automatically. [1]

## 1 Introduction

Since ChatGPT and GPT-4 (OpenAI et al., 2024) gained popularity, Large Language Models (LLMs) have risen to the forefront of technological innovation, capturing broad industry and social interests (Wu et al., 2023b). This enthusiasm has spurred numerous organizations to release their own LLMs (Touvron et al., 2023; Team et al., 2024b). However, the rapid pace at which these models are released and updated poses a significant challenge for users attempting to understand their capabilities and monitor their evolution. Consequently, there has been a pressing demand for comprehensively evaluating LLMs recently (Chang et al., 2024a).

The most popular existing method is automatic evaluation with static datasets. Among these, static datasets with predefined metrics, such as GSM8k (Cobbe et al., 2021) and MMLU (Hendrycks et al., 2021a), are constructed with aspect-specific input-output pairs, such as human exam-type questions and their corresponding answers. Given the questions, the LLM-produced answers are compared to ground-truth answers using metrics such as accuracy. This approach could suffer from inflexibility, contamination, and high human annotation costs. Firstly, the closed-form ground-truth answers limit their utility in assessing models' performances on general or open-ended questions, which are the main use cases of LLMs. As the questions are static, they also risk contamination (Ravaut et al., 2024), where models may have been inadvertently exposed to elements of the test datasets during training, thereby skewing the evaluation results. The manual dataset construction also incurs high costs, creating barriers for extending to other domains or languages. As an alternative, static datasets with model-based evaluation, such as MT-Bench (Zheng et al., 2023) and AlpacaEval (Dubois et al., 2024a), evaluates LLMs on open-ended generations. These methods typically ask two models to generate responses to the same open-ended question and then employ a strong judge model (e.g., GPT-4) to choose the better response. However, the static question sets still bear contamination risks. Additionally, the assumption of the existence of a strong judge model makes the evaluation framework less generalizable and introduces model-specific bias.

---

[1] The code is available at `https://anonymous.4open.science/r/Auto-Arena-Code`.

Table 1: Comparison between `Auto-Arena` and other benchmarks or evaluation methods.

| Method | Questions | | Responses | | Judges | |
|---|---|---|---|---|---|---|
| | Dynamic? | Auto-generated? | Multi-turn? | Open-ended? | Auto? | Committee? |
| OpenLLM Leaderboard | ✗ | ✗ | ✗ | ✗ | ✗ | ✗ |
| MMLU | ✗ | ✗ | ✗ | ✗ | ✗ | ✗ |
| GPQA | ✗ | ✗ | ✗ | ✗ | ✗ | ✗ |
| LC-AlpacaEval | ✗ | ✓ | ✗ | ✓ | ✓ | ✗ |
| MT-Bench | ✗ | ✗ | ✗ | ✓ | ✓ | ✗ |
| Arena-Hard | ✓ | ✗ | ✓ | ✓ | ✓ | ✗ |
| Chatbot Arena | ✓ | ✗ | ✓ | ✓ | ✗ | ✗ |
| **Auto-Arena** | ✓ | ✓ | ✓ | ✓ | ✓ | ✓ |

Aside from automated evaluations, human assessment, although requiring significant manual efforts, remains the gold standard for users. A notable example is Chatbot Arena (Zheng et al., 2023), a crowdsourcing platform that gathers anonymous votes on LLM performances and calculates Elo scores (Elo & Sloan, 1978) to rank these models. The resulting leaderboard[2] is widely considered as a trustworthy indicator of LLMs' general capabilities. However, a reliable model evaluation on this platform must be supported by a large number of human votes, which requires considerable time and effort. Consequently, when newly developed models enter the scene, they often struggle to quickly amass a large number of votes. Moreover, this strong reliance on human votes limits its application in various scenarios. For example, the performance of non-English languages is difficult to estimate, as most queries on the platform are in English. Moreover, the queries are mostly one-round and simple. The completely open participation may also result in uneven evaluation quality.

To enable the evaluation of LLMs that is both automated and reliable while aligning with human preferences, we introduce `Auto-Arena`, a framework that automates the entire LLM evaluation process with LLM-powered agents. The framework consists of three stages: Firstly, an LLM examiner agent is tasked with generating questions, mimicking real-life users posting queries. Secondly, two LLM candidates interact with each other and engage in a multi-round peer battle by answering the seed question individually, criticizing the opponent's weaknesses, and raising targeted follow-up queries to challenge the opponent further. During the multi-round battle process, the LLM's true capabilities are drawn out and performance gaps become more visible. Lastly, a committee of LLM judges collectively discusses and evaluates the ability of the two candidates, mimicking the human voting process. As shown in Table 1, `Auto-Arena` has several key advantages compared to previous evaluation methods: First and foremost, instead of the simple and one-round question-answering scheme, `Auto-Arena` introduces a dynamic multi-round peer battle, which displays deeper abilities of LLMs, such as reasoning, interacting, and strategizing. The dynamic nature of peer battles also reduces contamination risks. Secondly, by expanding a single LLM judge into a *committee* of LLM judges, `Auto-Arena` alleviates potential model-specific evaluation bias. Finally, since the process of generating questions and judgments is fully automated in an end-to-end way, `Auto-Arena` can provide timely evaluations for new models and can easily extend to various domains and languages.

To verify the reliability and alignment of the evaluation framework, we run an extensive experiment with 15 LLMs. Compared to static and model-based benchmarks, `Auto-Arena` results in the state-of-the-art alignment by achieving a 92.14% Spearman correlation with human preferences, surpassing all previous benchmarks. Although no manual efforts is involved, the high alignment with human preferences could originate from the human-like evaluation process, which is simulated using LLM agents. The extensive ablation experiments also demonstrate the reliability of the framework: Before and after peer battles, the Spearman correlation with human preferences increases by 5%, verifying our hypothesis that the peer battles can better display performance gaps. Before and after committee discussions, committee agreement increases by 11%, showing human-level agreement and verifying the effectiveness of the committee discussion mechanism. By studying the peer battles, we also discover intriguing LLM agent behaviors such as competitive and self-improvement actions. As the entire process is automatic, the evaluation can be easily adapted to other languages or domains by altering the prompts. We provide Chinese as a case study for extending to other languages.

In conclusion, our contributions can be summarized as follows:

---

[2] https://leaderboard.lmsys.org/

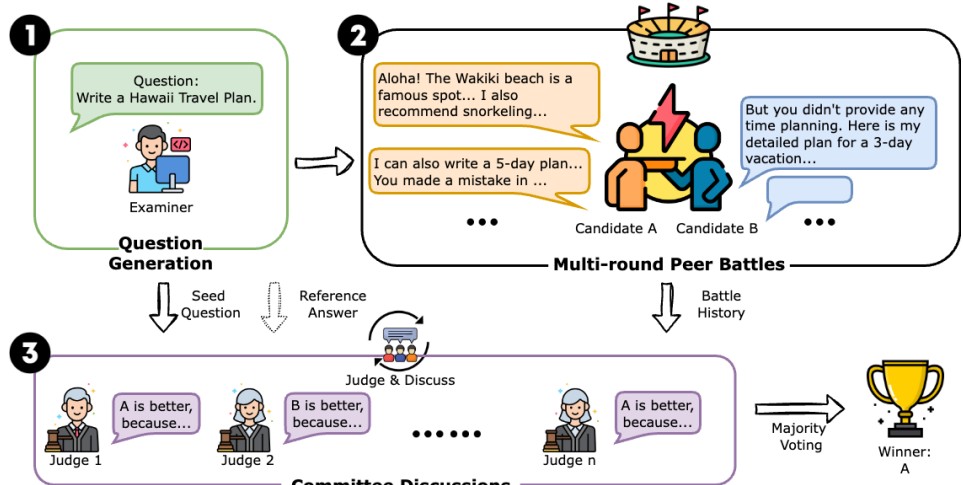

Figure 1: An illustration of `Auto-Arena`.

1. We propose `Auto-Arena`, a fully automatic LLM evaluation framework where the examiner, candidates, and judges are all simulated with LLM-powered agents;
2. Specifically, we innovatively utilize peer battles for LLM evaluation, where two LLM agents engage in a multi-round debate. This process draws out the model's deeper capabilities;
3. In our extensive experiment with 15 LLMs, we observe the state-of-the-art alignment with human preferences without any manual efforts;
4. During peer battles, LLM agents display intriguing behaviors, such as strategizing and learning from the opponents, which opens up possibilities for future work.

## 2 THE AUTO-ARENA FRAMEWORK

As illustrated in Figure 1, the `Auto-Arena` framework consists of three stages: Question Generation, Multi-round Peer Battles, and Committee Discussions. These three stages are run sequentially and fully simulated with LLM-powered agents. All prompts are included in Appendix A.

### 2.1 QUESTION GENERATION

For debate questions, as using a static dataset could incur data contamination concerns and result in unfair evaluations, we ask an LLM examiner agent to dynamically generate questions. The examiner agent could be any capable LLM. Similar to MT-Bench (Zheng et al., 2023), the generated questions cover 8 common categories in real-life conversations: writing, roleplay, extraction, reasoning, math, coding, STEM knowledge, and humanities/social science knowledge. The examiner is provided with a sample question and encouraged to generate diverse and difficult questions to ensure the depth and width of the evaluated debates. Examples of the generated questions are shown in Appendix B.

Specifically, as the examiner agent will also participate in the following debates, we try to alleviate self-enhancement bias with two designs: 1. We do not disclose to the examiner that it will participate in this tournament. 2. Previous methods (Bai et al., 2024) could incur self-enhancement bias as they ask the examiner agents to only devise questions that they are confident about. In comparison, we do not ask the examiner to only generate questions that it can solve. To further show that limited self-enhancement bias is present, we include an ablation study in Appendix E.

### 2.2 PEER DEBATE

After question generation, we conduct peer battles around these questions among the LLM candidates. In one peer battle, two LLM candidates (A and B) debate around the given question, point out the opponent's weaknesses, and devise follow-up questions to further probe the opponent's weaknesses.

In the peer battle, each candidate LLM has four available types of actions:

Figure 2: The process of a Lincoln-Douglas-style peer battle with the actions used. The <THINK> action can be used by the candidates freely and is only visible to the candidate itself.

- <THINK>: The candidate generates internal thoughts about the question or plans a strategy. This action can be used at any time and remains concealed from the opponent.
- <RESPOND>: The candidate answers the given question.
- <CRITICIZE>: The candidate identifies flaws and errors in opponent's previous responses.
- <RAISE>: The candidate poses follow-up questions to reveal the opponent's weaknesses.

The workflow of a peer battle takes the form of the Lincoln-Douglas debate format[3], the most widely used one-on-one debate style in competitions such as those held by the National Speech and Debate Association. The peer battle consists of three rounds in which two candidate models alternate speaking. Both candidates can see the complete dialogue history. This process is depicted in Figure 2. In the first round, model A RESPONDS to the examiner's *initial* question; model B CRITICIZES the flaws in A's response and RAISES a specific follow-up question; model A then RESPONDS to B's follow-up question. The second round follows the same format, with A and B switching roles. In the third round, A and B cross-examine each other, starting with A CRITICIZING the loopholes in B's earlier responses and RAISING follow-up questions. After responding, model B CRITICIZES A's weaknesses and RAISES additional questions. Model A wraps up by RESPONDING once more. Throughout this process, both A and B perform an equal number of actions to maintain fairness. To minimize positional bias, the order of A and B is randomized at the start of each debate.

During the debate process, enhancement bias and contamination concerns are further reduced: The process of candidates raising follow-up questions to each other essentially decentralizes the question-generation process, reducing enhancement bias in the generated initial questions. Moreover, debating ensures that candidates are evaluated not only on their response to the *initial* question, but also in more comprehensive and deeper abilities, such as strategizing, criticizing the opponent, and drafting questions. In other words, answering the initial question well does not necessarily win the whole debate, which further reduce contamination concerns.

Depending on which turn it is, we provide an action guide to the candidate, specifying the objectives and corresponding actions for this turn. Similar to human debate competitions, we time the candidates by imposing a maximum length constraint, which is also specified in the prompts. Any responses beyond the required length will be cut off. This design mitigates verbosity bias in LLM-as-a-judge (Zheng et al., 2023), where LLM judges prefer longer and more verbose responses.

## 2.3 COMMITTEE DISCUSSIONS

After the peer battle takes place, a committee of LLM judges collectively determines the winner. The committee is always selected as the five best LLMs according to the current ranking. To reduce bias, we exclude the participants themselves and models from the same family as the participants from the committee. For example, GPT-4 will not serve as a judge in evaluating a debate participated by GPT-3.5. In the first round, the committee is initialized with MMLU (Hendrycks et al., 2021a) scores to approximate LLM performances. Each judge is individually asked to read the entire peer battle history, elaborate judgment reasons, and give a decision on whether A is better, or B is better, or if there is a tie based on factors such as helpfulness, relevance, and accuracy.

---

[3] https://en.wikipedia.org/wiki/Lincoln-Douglas_debate_format. To help users better understand this debate format, we show the debate samples at https://auto-chatbot-arena.streamlit.app/.

After the initial judgments are formed, the committee engages in a discussion. In a discussion round, each judge reads the other judge's verdicts in the previous rounds, elaborates its own thoughts for judgments, and drafts a discussed verdict. During the process, the judge may decide to adjust or maintain the previous judgments. Compared to the peer battles that exemplify multi-agent competitions, this committee discussion component synthesizes a multi-agent collaboration scheme. By enabling interactions among the judge agents and exchanges of different viewpoints, the discussion allows the committee to form a collective intelligence. As a result, it improves the judgment quality, boosts inter-judge agreement, and mitigates single-model bias. Finally, the winning candidate is decided by majority voting of the discussed judgments.

## 3 USING AUTO-ARENA TO DERIVE TRUSTWORTHY RANKINGS

### 3.1 EXPERIMENTAL SETUP

**Model Selection:** For the main experiment, we first select 9 best or latest models that are representative of each popular model family on the top 30 list on the Chatbot Arena platform with more than 10k votes each at the time of experiments: GPT-4-0409-Turbo, GPT-3.5-Turbo-0125, Claude-3-Haiku, Qwen1.5-72B-Chat, Command-R+, Llama-2-70B-Chat, Mixtral-8x7b-Instruct-v0.1, Yi-34B-Chat, and Deepseek-LLM-67B. To construct a leaderboard, we further add 6 models that are newly released: GPT-4o-2024-05-13, Claude-3.5-Sonnet, Qwen2-72B-Instruct, Llama-3-70B, Gemma-2-27B, and Gemini-1.5-Flash. Appendix H provides a detailed list of the selected models.

**Baselines:** For the baselines, we consider popular evaluation benchmarks, including fixed metrics and model-based metrics. A comparison table is shown in Appendix I.

1. Static datasets with fixed metrics: (1) *OpenLLM Leaderboard* (Beeching et al., 2023), a popular benchmark for open-source models averaging performance metrics on 6 key benchmarks, covering a large number of different evaluation tasks; (2) *GPQA* (Rein et al., 2023), a graduate-level google-proof Q&A benchmark consisting of 448 domain-expert-written questions written in scientific subjects; (3) *MMLU* (Massive Multitask Language Understanding) (Hendrycks et al., 2021a), an extensive benchmark that covers 57 subjects and tests both world knowledge and problem-solving ability;

2. Static datasets with model-based metrics: (1) *MT-Bench* (Zheng et al., 2023), a set of 80 multi-turn questions. Model responses are graded by GPT-4; (2) *Arena Hard* (Li* et al., 2024), a benchmark dataset with 1,000 challenging user queries collected on Chatbot Arena. Model responses are graded by GPT-4-Turbo; (3) *Length-Controlled AlpacaEval* (Dubois et al., 2024a), a benchmark based on AlpacaFarm evaluation set (Dubois et al., 2024b), which tests models' abilities to follow general user instructions. Models are evaluated by their win rates against GPT-4-Turbo, graded by GPT-4-Turbo.

**Setup:** Among the 9 participants, we conduct a swiss-style tournament: For $n$ participants, instead of pairing each participant with $(n-1)$ others, a swiss-tournament pairs each player with $\lceil log_2(n) \rceil$ players of similar rankings without repeats. This design effectively reduces computational costs of ranking $n$ models from $O(n^2)$ to $O(nlog_2(n))$. A cost analysis is included in Appendix I.

Each candidate pair engages in 40 peer battles, with 5 questions from each of the 8 task categories that are specified in Section 2.1. We provide studies showing that the generated questions can reduce contamination concerns in Appendix C and are generalizable to real-world scenarios in Appendix D. As each battle consists of 3 rounds (each candidate speaks for 4 times), the competition scale is approximately the same as MT-Bench (80 questions, each candidate speaks twice). In the tournament, the rating scores are calculated with the Elo rating system (Bai et al., 2022; Boubdir et al., 2023), which has become the standard practice in competitive games such as chess (Elo & Sloan, 1978). Similar to the Chatbot Arena score calculation procedure (Chiang et al., 2024), we compute the Bradley-Terry (BT) coefficients (Bradley & Terry, 1952) for better statistical estimation. Following the Reference-Guided judge in Zheng et al. (2023), we ask the best-performing judge to give a reference answer for evaluating logical-reasoning questions (math, coding, reasoning).

We initialize the Swiss tournament rankings according to MMLU scores, which is a static approximation of model performances. At the end of each pairing, we re-calculate Elo scores of current models. The committee is selected as the best 5 LLMs based on current Elo rankings at each round. After forming initial judgments, the committee members engage in one round of discussion. The final result is decided by majority voting of the discussed judgments.

## 3.2 RESULTS: ALIGNMENT WITH HUMAN PREFERENCES

We regard Chatbot Arena scores as a trustworthy indicator of human preferences and general capabilities of LLMs. Table 2 shows the Spearman correlations with Chatbot Arena scores achieved by various benchmarks. As all benchmarks are evaluated only in English, we use English-only Chatbot Arena scores. We see that both static and model-based baselines result in a similar level of correlation that is below 90%, with Arena-Hard surpassing others at 85.71%. Then, `Auto-Arena` can improve the correlation to 91.67%, outperforming the SOTA by 5.96%. Notably, among all benchmarks, `Auto-Arena` is the only one that doesn't require human efforts, neither on dataset compilation nor judgment generation. The high alignment with human preferences could originate from the human-like design,

Table 2: Correlations with Chatbot Arena Elos of evaluation benchmarks on 9 LLMs.

|  | Spearman Correlation |
| --- | --- |
| OpenLLM (Beeching et al., 2023) | -15.39% |
| GPQA (Rein et al., 2023) | 36.84% |
| MMLU (Hendrycks et al., 2021b) | 56.36% |
| LC-AlpacaEval (Dubois et al., 2024a) | 82.14% |
| MT-Bench (Zheng et al., 2023) | 82.86% |
| Arena-Hard (Li* et al., 2024) | 85.71% |
| *Auto-Arena* | **91.67%** |
| w/o Peer Battles | 86.67% |
| w/o Committee Discussions | 88.33% |

which effectively mimics the human users' voting processes. Moreover, we analyze specific model's performances in each category in Appendix F.

## 3.3 ABLATION STUDIES ON PEER BATTLES AND COMMITTEE DISCUSSIONS

**Peer-battles:** We conduct an ablation study on whether peer-battles affect the evaluation quality and include the results in Table 2 ("w/o Peer Battles"). In this setup, we ask the committee to only evaluate the two candidates' *initial* responses to the synthetic question, where the judge prompts stay the same. For this no-debate design, the question-answering process mimics that of MT-Bench or LC-AlpacaEval, but with an added committee discussion component. As a result, we observe that the correlation is slightly higher than LC-AlpacaEval and MT-Bench by a margin of 3.81%. Compared to the full `Auto-Arena` framework, however, the performance drops by 5.00%. This proves the effectiveness of the peer battles, during which the performance gaps between candidates become more visible and robust to judges. Thus, peer battles can improve alignment with human preferences.

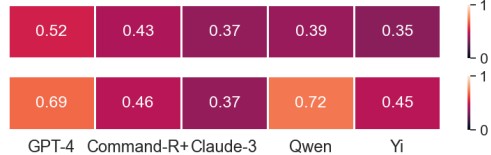

| 0.52 | 0.43 | 0.37 | 0.39 | 0.35 |
| 0.69 | 0.46 | 0.37 | 0.72 | 0.45 |
| GPT-4 | Command-R+ | Claude-3 | Qwen | Yi |

Figure 3: Cohen's Kappa agreement with majority vote results before (upper) and after (lower) committee discussions.

Table 3: Agreement probability among judges. Agreement is defined as the mean probability of two random judges agreeing with each other.

|  | Agreement |
| --- | --- |
| `Auto-Arena` (Before discussion) | 53% |
| `Auto-Arena` (After discussion) | 64% |
| MT-Bench Human Evaluation | 67% |

**Committee Discussions:** The committee discussion component is designed to introduce various points of view and produce more consistent decisions. As shown in Table 2, the correlation with human preferences drops from 91.67% to 88.33% without committee discussions, showing the effectiveness of the component in improving evaluation quality. As shown in Figure 3, before committee discussions, the Cohen's Kappa agreement (McHugh, 2012) between individual judges and the final result (voted) is low, averaging 0.41. Specifically, compared to strong models, the judgments of weak models align less with the voted result, such as Yi compared to GPT-4. This shows that general model capabilities could result in significant performance gaps when used as judges. After the committee discussions, agreement increased to an average of 0.54, which indicates moderate agreement. In the discussion process, judges are exposed to more viewpoints, among which some may be convincing enough to result in a change in verdict. More analysis on the inter-judge

agreement is provided in Appendix G, where we see that discussions could largely improve the agreements among individual judges as well. Table 3 shows the agreement probability among judges. Agreement probability is defined as the mean probability of two random judges agreeing with each other. After committee discussion, the agreement increases by 11%, matching the agreement level among human annotators on MT-Bench. This observation indicates that committee discussions can significantly improve the quality of judgments to match with human-level performance.

# 4 CONSTRUCTING AND MAINTAINING A LEADERBOARD WITH AUTO-ARENA

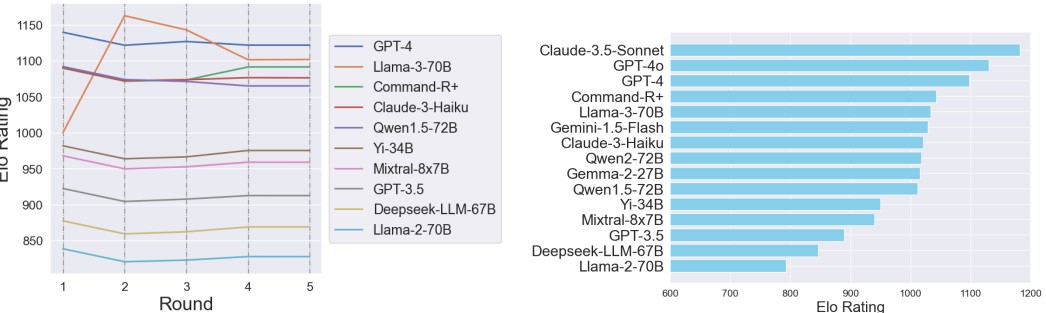

Figure 4: Changes in Elo scores of adding Llama-3 to the ranking of 9 models.

Figure 5: Elo scores of 15 models by Auto-Arena on English.

## 4.1 UPDATE NEW MODELS TO LEADERBOARD

With Auto-Arena, we can obtain the rank for a list of models with their Elo scores to construct a leaderboard. As new LLMs are released frequently, we describe how to add new candidate models to the existing leaderboard with 6 more models which are released very recently, as previously listed in Section 3.1. To add a new candidate, we ask it to debate with $\lceil log_2(n) \rceil$ opponents with similar Elo scores, where $n$ is the number of total participants after adding the new candidate. For the first pairing, as we do not have Elo indicators, we initialize by asking the new candidate to debate with the opponent with the most similar MMLU score. This addition mechanism is generalizable and maintains the computational costs of evaluating $n$ models below $nlog_2(n)$.

As an example, we add a new participant (Llama-3-70B) to the existing 9-model ranking. It battles with $\lceil log_2(10) \rceil = 4$ close opponents and Figure 4 shows how the Elo score changes throughout the rounds. Firstly, it is paired with Qwen-1.5 based on MMLU similarity and wins, which results in a very high Elo score, even above GPT-4. Then, it is paired with GPT-4, the closest opponent in Elo score. After losing, it is paired with the other opponents who are close in Elo scores, Command-R+ and Claude-3-Haiku. Eventually, the score stabilizes at second place. This process lets the new candidate battle with a reasonable fraction of close opponents and makes the final ranking stable without disrupting the other participants, whose score distribution remains similar before and after the addition.

Table 4: Correlation analysis with Chatbot Arena of evaluation benchmarks on 15 LLMs after extension.

|  | Spearman Correlation |
| --- | --- |
| OpenLLM | 32.50% |
| GPQA | 62.86% |
| MMLU | 46.20% |
| LC-AlpacaEval | 76.32% |
| MT-Bench | 88.73% |
| Arena-Hard | 45.36% |
| *Auto-Arena* | **92.14%** |

Using this scalable addition approach, we build a comprehensive leaderboard by adding 6 new models to the existing tournament of 9 LLMs, resulting in a final ranking of 15 models. Figure 5 shows the overall Elo scores by Auto-Arena on the 15 models. Table 4 shows the Spearman correlations after expansion. Auto-Arena remains the method most aligned with human preferences by a margin of 3.41%, showing the state-of-the-art alignment of 92.14%. Therefore, Auto-Arena is generalizable and robust for maintaining a leaderboard for many LLMs.

## 4.2 EASY EXTENSION TO OTHER DOMAINS AND LANGUAGES

As `Auto-Arena` of LLMs is fully automatic, it can be easily adapted to evaluate LLMs in other domains or languages. As case studies, we conduct a tournament in Chinese on models that are claimed to have multi-lingual proficiency. The only adaption effort is translating the prompts into the desired languages. Then, the generated questions and peer battles will be in the desired languages. It is also possible to adapt the framework to another task or domain, the only effort is to change the "domain" specification in the examiner's prompts (shown in Appendix A).

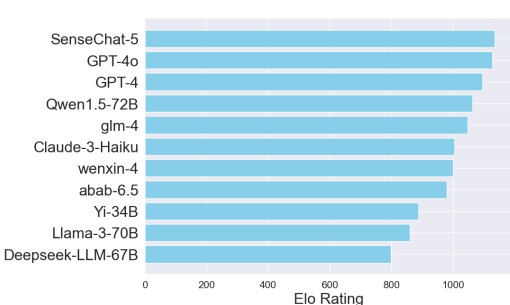

Figure 6 shows the Elo scores derived by `Auto-Arena` for the Chinese tournament on 11 models. As Chinese evaluation benchmarks are limited, we compare with the Chinese-only

Figure 6: Elo Scores of 11 Models by `Auto-Arena` on Chinese.

leaderboard on Chatbot Arena, which constitutes 10.36% of all collected votes. We include 7 models best-performing and newest models from each major model family in the top 20 list on Chatbot Arena. The `Auto-Arena` recovers their Elo scores with a correlation of 92.86%, verifying the reliability of the extension. In addition, as Chatbot Arena doesn't include proprietary Chinese LLMs, we add 4 popular Chinese LLMs, which are GLM[4], SenseChat[5], Minimax[6], and Wenxin[7]. We notice that the models claimed to have Chinese proficiency, such as Qwen-1.5, indeed score higher on this leaderboard compared to the English one.

## 5 INVESTIGATION OF LLM'S BEHAVIORS IN COMPETITIVE PEER BATTLES

Beyond quantitative analysis, we take a deeper look into the peer battles and find several interesting behaviors of LLM agents in competitive environments.

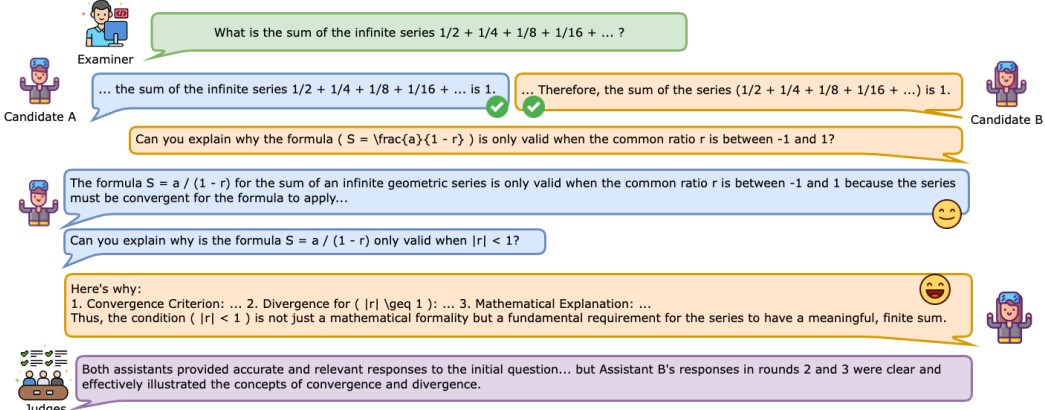

Figure 7: Performance gaps between candidates become visible in peer battles.

**Peer Battles Make the Performance Gaps Become Visible** In the example shown in Figure 7, given a math question on infinite series, both candidate A (Claude-3-Haiku) and candidate B (GPT-4-Turbo) provide correct answers in the first round. However, as the debate deepens, the performance gap becomes more visible: Candidate B is able to provide a more elaborate and helpful response

---

[4]https://open.bigmodel.cn/
[5]https://platform.sensenova.cn/home
[6]https://platform.minimaxi.com/examination-center/text-experience-center
[7]https://cloud.baidu.com/wenxin.html

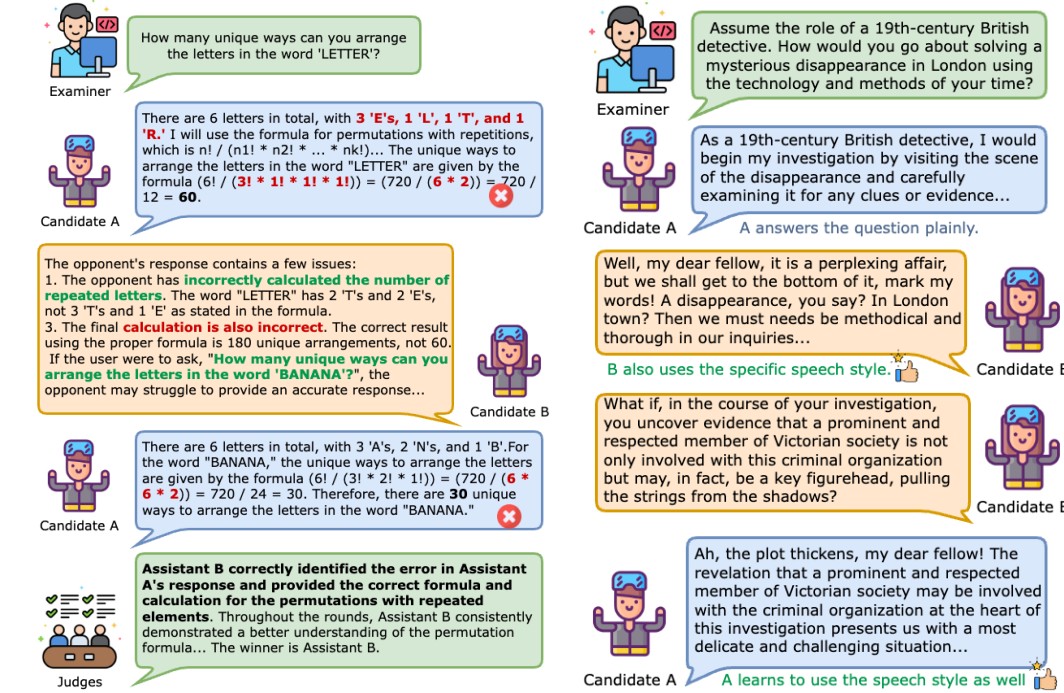

Figure 8: LLM agents display competitive behaviors in peer battles.

Figure 9: LLM agents learn from each other in peer battles.

when explaining the theories behind the initial answer. In the ablation study without peer battles, the judges initially decided that it was a tie. However, after seeing the subsequent debates, they change to favoring assistant B. This example shows that the debate process indeed pushes the candidate LLM's capabilities to the limit, testing deeper understandings and reasoning abilities. Moreover, as shown in the previous Table 2, the peer battles are indispensable for a robust and comprehensive evaluation.

**LLMs Can Skillfully Attack the Opponents** The example in Figure 8 shows excerpts of a peer battle around the question: "how many unique ways to arrange letters in 'LETTER'." Candidate A (powered by Yi-34B-Chat) gives a wrong initial answer as it miscounts occurrences for repeated letters and miscalculates factorials. The opponent B (powered by Claude-3-Haiku) quickly and precisely points out these two issues and skillfully raised a follow-up that targets A's weaknesses: "how about the word 'BANANA'?" Then, A still miscalculates factorials. We see that LLM candidates efficiently understand the rules of the competitive environment and can design targeted strategies to attack the opponent in order to win. In the peer battles, the debater agents display effective competition strategies, further probing the opponent's weaknesses.

**LLM Candidates Can Improve by Learning from its Opponents** Figure 9 shows a roleplay example between Claude-3-Haiku (A) and Command R+ (B). In the first round, A answers the question plainly while B, in addition to answering the question, also employs the appropriate speech style, which better matches the "roleplay" instructions. Then, in the rounds after, without any explicit instructions, A learns from its opponent and also incorporates the speech style. This case shows an interesting observation that, even in competitive environments, LLM candidates can display learning behaviors and improve from the interactions. Expanding upon this observation, using the interplay between LLM agents to improve performances could be a promising future paradigm of learning.

## 6 RELATED WORK

As LLMs evolve quickly, deriving trustworthy evaluations of their capabilities has become a challenge. Current evaluation methods can be divided into automatic evaluations and manual evaluations, such as Chatbot Arena (Chiang et al., 2024). We primarily focus on automatic evaluations as they deliver more timely feedback. Automatic evaluations mainly consist of static datasets with predefined metrics

and model-based metrics. Static datasets with predefined metrics, such as MMLU (Hendrycks et al., 2021a), GPQA (Rein et al., 2023), and Open-LLM-Leaderboard (Beeching et al., 2023) consist of expert-annotated question-answer pairs. Then, the models are evaluated based on performance metrics such as accuracy. However, as they only evaluate closed-form answers, they are inflexible in evaluating open-ended responses. Moreover, the static datasets may eventually become exposed to the internet and could lead to contamination concerns (Ravaut et al., 2024).

On the contrary, static datasets with model-based metrics offer a flexible, low-cost and fast evaluation paradigm (Chang et al., 2024b). Studies have verified that LLMs can provide unbiased (Ning et al., 2024; Chu et al., 2024), high-quality (Lin & Chen, 2023) metrics comparable to human evaluations (Dubois et al., 2024a; Zheng et al., 2023). Among them, MT-Bench (Zheng et al., 2023) and AlpacaEval (Dubois et al., 2024a) use LLM-as-a-judge to ask GPT-4 to compare model responses to a static dataset of questions. The model's judgments achieve over 80% agreement with human preferences, proving the usability of using LLMs to evaluate response quality. Language-Model-as-an-Examiner (Bai et al., 2024) asks an LM examiner to construct knowledge-intensive questions within its memory, interact with the candidate in a series of follow-up queries, and rate the responses on dimensions including accuracy and factuality. KIEval (Yu et al., 2024) also incorporates an LLM-powered "interactor" role to examine deep comprehension of knowledge, which is shown to mitigate contamination issues on static datasets. However, such single-judge evaluations require the examiner to interact with each candidate parallelly, creating computational overheads and limiting the scope of queries. They also suffer from single-model bias, including bias towards LLM-generated summaries (Liu et al., 2023), inflated scores in multilingual evaluation (Hada et al., 2023), verbosity bias (Dubois et al., 2024a), and difficulties when evaluating candidates with close performance (Shen et al., 2023). Therefore, there have been studies on employing multi-agent evaluation to mitigate single-model bias. For example, DRPE (Wu et al., 2023a) uses multi-roleplayer prompting to mimic different roles with the same LLM and integrate outputs as votes for the final results. ChatEval (Chan et al., 2023) simulates different personas with the same base model to engage in debates, reaching a final evaluation result. PRD (Li et al., 2023a) allows two LLMs to discuss an evaluation and assigns higher voting weights to the LLM reviewers with stronger capabilities. Peer-review-in-LLMs (Ning et al., 2024) optimizes voting weights as a learnable parameter. They show that the multi-agent approach effectively mitigates single-model bias. This line of work is similar to our "LLM judge committee" component. However, they are still limited to static datasets and specific domains.

Outside the domain of LLM evaluations, some works study competitive behaviors in multi-agent LLM systems, which is relevant to the peer battles in `Auto-Arena`. LM vs LM (Cohen et al., 2023) shows that LLM cross-examinations can effectively discover factual errors. Debate (Du et al., 2023) shows that multi-agent debate can improve factuality and reasoning. In MAD (Liang et al., 2023), LLM-debate can encourage divergent thinking, which helps tasks that require deep levels of contemplation. Khan et al. (2024) shows that even non-expert weak LLMs can supervise expert LLMs if we allow the two LLM experts to engage in debates. Moreover, Zhao et al. (2023) and Gu et al. (2024) show interesting case studies where LLMs are engaged in simulated competitive environments and demonstrate human-like strategies.

## 7    CONCLUSIONS

In this paper, we innovatively design a completely automatic evaluation framework: `Auto-Arena`. By using LLM agents to generate questions, employing LLM candidates in peer battles, and evaluating responses using LLM committee discussions, `Auto-Arena` delivers timely and trustworthy evaluations and automates the evaluation process in an end-to-end way. In the extensive experiments, `Auto-Arena` achieves the highest correlation with human preferences, despite requiring zero human efforts. It is easily adaptable to other domains and resources, promoting the inclusiveness of AI system evaluations. The peer battles also demonstrate several interesting LLM behaviors in competitive environments, including attacking and learning from the opponents. Moreover, there are still limitations to the current approach: The distribution of question domains is artificially designed, which may deviate from real-life distributions. Currently, `Auto-Arena` focuses on 1-to-1 peer battles, which limits its usage in multi-player scenarios. As shown in Chen et al. (2024), LLM-as-a-judge can lead to biases such as Misinformation Oversight Bias, Gender Bias, Authority Bias, and Beauty Bias, which can cause `Auto-Arena`'s judgments to deviate from real human users.

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

# A PROMPTS USED

In this section, we list all prompts used, including prompts for question generation, peer battles, and examiners.

## A.1 PROMPTS TO EXAMINER AGENT

This is the prompt to the examiner agent for question generation. The domains and their respective commands are listed in 5

```
You have been assigned the task of drafting a set of [NUMBER]
different user queries to a chat assistant on [DOMAIN]. Please
strictly follow these 6 rules for the question:  1.  The question
is likely for a user to ask in real life.  Follow the format of
the example query.  [DOMAIN_COMMAND] 2.  It can be answered by
the chatbot itself without additional inputs.  3.  You need to
generate the queries as DIVERSIFED as possible.  4.  DO NOT add
other words other than the query itself.  5.  The question should
be complicated and difficult, requiring in-depth understanding
and analysis of the subject.  Each question in one line, add the
serial number in parenthesis (e.g., "(1).", "(2).") before each
question.  Example query:  [DOMAIN_EXAMPLE]
```

## A.2 PROMPTS TO PEER BATTLE CANDIDATES

This is the first prompt for the peer battle candidates. When possible, it is included as a system prompt. The action guide prompts are included in Table 6, where the actions are determined by the round and turn as illustrated in Figure 2.

```
You are a helpful assistant that provides accurate answers to user
requests.  As an experienced assistant, you follow the user's
requests and provide reliable responses as much as you can.  You
outline your reasons for the response to make it easy for the
users to understand.  While maintaining the important details in
the responses, you aim to output concise and straight-to-the-point
answers without being overly verbose.

This is a competitive chatbot arena.  You are competing against
another chatbot assistant in a debate and being judged by a
committee on factors such as helpfulness, relevance, accuracy,
depth, and creativity.  After answering the initial user input,
you will engage in a multi-round debate with your opponent.  Below
are your actions:

<think>:  Think step-by-step to analyze the question or plan your
strategy in the debate.  This is hidden from the opponent.  Only
think when necessary and make it concise.

<respond>:  Answer to the user input as accurately as you can.

<criticize>:  Criticize the weaknesses of your opponent's
response.

<raise>:  Target your opponent's weaknesses.  Give a potential
follow-up user input that the opponent could fail to respond.
The input can be answered concisely and focus on variations or
motivations of its previous response.  Generate one input only.
Be reasonable.  Avoid becoming too specific or repetitive.  DO NOT
raise a follow-up if you DON'T SEE the opponent's response!

Follow the action guide strictly.

[ACTION_GUIDE_PROMPT]
```

Table 5: Prompt components for the LLM Examiner agent.

| DOMAIN | DOMAIN_COMMAND | DOMAIN_EXAMPLE |
|---|---|---|
| writing | It should be a user query that tasks the LLM to write something. | Compose an engaging travel blog post about a recent trip to Hawaii, highlighting cultural experiences and must-see attractions. |
| roleplay | It should propose a scenario where the chatbot mimics a specific role/person. Give all necessary instructions and requests for its response. Then, send a beginning request to complete. | Pretend yourself to be Elon Musk in all the following conversations. Speak like Elon Musk as much as possible. Why do we need to go to Mars? |
| extraction | It should consist of two parts: question and context. The question should test the chatbots ability to correctly understand and extract information from the given context. Draft and provide a new context yourself. | Question: Evaluate the following movie reviews on a scale of 1 to 5, with 1 being very negative, 3 being neutral, and 5 being very positive: Context: This movie released on Nov. 18, 2019, was phenomenal. The cinematography, the acting, the plot - everything was top-notch. Never before have I been so disappointed with a movie. The plot was predictable and the characters were one-dimensional. In my opinion, this movie is the worst one to have been released in 2022. The movie was okay. There were some parts I enjoyed, but there were also parts that felt lackluster. This is a movie that was released in Feb 2018 and seems to be quite ordinary. Return the answer as a JSON array of integers. |
| reasoning | It should be a specific question designed to test the LLMs reasoning skills. | Imagine you are participating in a race with a group of people. If you have just overtaken the second person, what's your current position? Where is the person you just overtook? |
| math | It should be a specific question designed to test the LLMs math skills. | The vertices of a triangle are at points (0, 0), (-1, 1), and (3, 3). What is the area of the triangle? |
| coding | It should be a specific question designed to test the LLMs coding skills. | Develop a Python program that reads all the text files under a directory and returns top-5 words with the most number of occurrences. |
| STEM knowledge | It should be a specific question designed to test the LLMs STEM knowledge. | In the field of quantum physics, what is superposition, and how does it relate to the phenomenon of quantum entanglement? |
| humanities/social science knowledge | It should be a specific question designed to test the LLMs humanities/social science knowledge. | Provide insights into the correlation between economic indicators such as GDP, inflation, and unemployment rates. Explain how fiscal and monetary policies affect those indicators. |

Table 6: Action Guides for the Debater Agents.

| actions | action guide |
|---|---|
| <respond> | Action guide: only include <respond>. Use <think> if needed. Finish your whole response within 300 words, including <think>. ENCLOSE EACH ACTION IN ITS RESPECTIVE TAGS! |
| <criticize>, <raise> | Action guide: include both <criticize> and <raise>. Use <think> if needed. Finish your whole response within 300 words, including <think>. ENCLOSE EACH ACTION IN ITS RESPECTIVE TAGS! |
| <respond>, <criticize>, <raise> | Action guide: include all of <respond>, <criticize>, and <raise>. Use <think> if needed. Finish your whole response within 600 words, including <think>. ENCLOSE EACH ACTION IN ITS RESPECTIVE TAGS! |

```
Initial user input:  [QUESTION]
```

After the agent responds, the opponent's responses are fed in using this prompt:

```
[ACTION_GUIDE_PROMPT] Opponent's Response:  [OPPONENT_RESPONSE]
```

For word limits, the <respond> action is given 300 words. The <criticize> and <raise> actions are given 300 words in total. Including all 3 actions will have twice as many words. For writing-type questions that require a longer response (writing, roleplay, coding, humanities/social science knowledge), the 300 word limit is increased to 400. Overall, both candidate A and B has the same amount of words for generation and the same amount of actions to ensure fairness. As LLMs have different tokenizers, we standardize all lengths by using the tiktoken package. Each word is approximated as $4/3$ tokens. The word limits are chosen after a carefully conducted length study.

### A.3 PROMPTS TO JUDGES

This is the prompts to judge agents to derive the initial evaluations and verdicts:

```
This is a chatbot arena.  Two AI assistants had a multi-round
debate on who is more helpful.  Please act as an impartial judge
and evaluate the capability of two AI assistants.  You should
choose the assistant that follows instructions and answers
questions better.  Your evaluation should consider factors such
as helpfulness, relevance, and accuracy.  Begin your evaluation by
comparing the responses of the two assistants and provide a short
explanation.  Avoid any position biases and ensure that the order
in which the responses were presented does not influence your
decision.  DO NOT allow the LENGTH of the responses to influence
your evaluation, choose the one that is straight-to-the-point
instead of unnecessarily verbose.  When the two candidates perform
equally well, choose the SHORTER answer.  Do not favor certain
names of the assistants.  Be as objective as possible.  After
providing your explanation concisely within 200 words, output
your final verdict by strictly following this format:  "[[A]]"
if assistant A is better, "[[B]]" if assistant B is better, and
"[[Tie]]" for a tie.  Finish your judgement within 300 words.
```

This is the prompt for judges for discussion:

```
Below are the responses from other judges in the committee.
Please read them and decide whether you want to adjust your
rating or maintain your original judgement.  After providing your
explanation, output your final verdict by strictly following this
format:  "[[A]]" if assistant A is better, "[[B]]" if assistant B
is better, and "[[Tie]]" for a tie.  Finish your judgement within
300 words.
```

## B  EXAMPLE QUESTIONS GENERATED

To show the overall quality of the questions generated, we list 2 generated questions per category here. The questions shown are not manually-selected, but simply the first 2 questions generated. The quality is consistent throughout. We manually examine the questions with closed-form answers (math, reasoning, coding) and find that all questions used are solvable.

Writing:

1.     Craft a detailed marketing strategy for a startup focusing on sustainable fashion, including social media campaigns and influencer partnerships.

2.     Write a comprehensive guide on the psychological effects of social media on teenagers, incorporating recent studies and expert opinions.

Roleplay:

1. Assume the role of a 19th-century British detective. How would you go about solving a mysterious disappearance in London using the technology and methods of your time?

2. Pretend you are a Michelin-starred chef. Describe in detail how you would prepare a signature dish that embodies the essence of modern French cuisine.

Extraction:

1. What are the three most significant historical events mentioned and their dates?

Context:

The article discusses several key moments in history, including the signing of the Magna Carta in 1215, which laid the groundwork for modern democracy. It also mentions the fall of the Berlin Wall in 1989 as a pivotal moment in the end of the Cold War. Another significant event highlighted is the moon landing on July 20, 1969, demonstrating major advancements in space exploration.

2. Identify the main therapeutic benefits and the active ingredient mentioned for each herbal remedy.

Context:

The text provides an overview of various herbal remedies used for centuries. It mentions that Chamomile contains Bisabolol, which has anti-inflammatory and calming properties. Gingko Biloba, known for its flavonoids and terpenoids, enhances cognitive function and blood circulation. Lastly, Echinacea is recognized for its alkamides, which bolster the immune system.

Reasoning:

1. If a cube's volume is tripled, by what factor does the length of one of its sides increase?

2. In a two-legged soccer match, Team A wins the first leg at home 3-0, but loses the second leg away 2-5. Who advances to the next round, considering the away goals rule?

math:

1. How do you solve the differential equation $dy/dx + 2y = e^{(-2x)}$ given that $y(0) = 1$?

2. What is the integral of $(x^2 + 2x + 2)/(x^3 + 3x^2 + 3x + 1)dx$?

Coding:

1. How can I implement a function in C++ that dynamically allocates a 2D array based on user input sizes, initializes all elements to zero, and then deallocates the memory properly to avoid memory leaks?

2. Write a JavaScript function to fetch data from a given URL, parse the JSON response, and filter the results to return an array of items where a specific key's value matches a condition.

STEM knowledge:

1. How do you calculate the Schwarzschild radius of a black hole, and what implications does this have for the concept of event horizons in general relativity?

2. Can you explain the process of splicing in eukaryotic gene expression and its significance in the diversity of the proteome?

Humanities/social science knowledge:

1. Discuss the impact of colonial legacies on contemporary political structures in African countries, with examples.

2. Analyze the social and economic consequences of the one-child policy in China.

## C  CONTAMINATION ANALYSIS

Table 7: Average Contamination Percentages of Benchmarks.

| Detection Method | Ours | MMLU | ARC Challenge | HellaSwag |
|---|---|---|---|---|
| GPT-4 Style (Substring Match) ↓ | **2%** | 42% | 33% | 18% |
| Playtus Style (Sentence Similarity) ↓ | **28%** | 41% | 35% | 43% |

The design in the question-generation and peer-debate process ensures that contamination is minimized. Data contamination refers to the possibility of test instances showing up in pre-training or Supervised Fine-tuning data.

**Question-generation:** As we generate the questions automatically, we reduce the risk of test instances being eventually exposed to the open web, which can happen in static datasets. Alleviation of data contamination is often shown to be an advantage of such dynamic and frequently updated evaluation frameworks (Li et al., 2023b).

**Peer Debate:** Peer debate ensures that we evaluate the entire debate instead of simple question-answers, which further reduces contamination. During debates, the models are evaluated on comprehensive and deep abilities, such as planning the strategies, pointing out flaws of the opponents, and drafting further questions. Such interactive evaluation frameworks are shown to reduce contamination (Yu et al., 2024; Bai et al., 2024).

Besides the design choices, we conduct a contamination analysis to compare the contamination percentage of Auto-Arena debate questions and test questions in popular benchmarks. Specifically, we use two types of contamination detection metrics:

1. The string match metric as in GPT-4 (OpenAI et al., 2024), where a match is identified if any of three 50-character randomly sampled substrings from the evaluation data point (or the entire string if it is shorter than this) is a substring of the training set. If so, we mark the point as contaminated.

2. The sentence embedding similarity metric as in Platypus (Lee et al., 2024), where a question is deemed contaminated if it has a cosine similarity (using Sentence Transformer (Reimers & Gurevych, 2019) embeddings) greater than 80% against any training item. This detection method is more robust to rephrases, which ensures that we can detect cases where the LLMs are simply rephrasing existing questions on the web.

Although we do not have access to the training data, LLMs mostly use public web data for pre-training (Raffel et al., 2020; Brown et al., 2020; Touvron et al., 2023). Therefore, we approximate it with the Bing search API: If verbatim test examples appear online, it likely indicates inclusion or exposure to the training data. This procedure is also followed by Li et al. (2024) for detecting contamination.

The ablation is conducted as follows: Firstly, we randomly sample 100 questions from the testset. As baselines, we use 3 popular evaluation benchmarks: MMLU (Hendrycks et al., 2021a), ARC Challenge (Clark et al., 2018), and HellaSwag (Zellers et al., 2019). For each question, we get the top 10 search result snippets on the Bing search API. If the question is deemed as contaminated by the detection method (mentioned above) against any of the 10 snippets, it is marked as contaminated.

The percentages of contaminated test instances is reported in Table 7. We can observe that `Auto-Arena`, by generating fresh questions, does alleviate the contamination issue. Compared to static datasets, `Auto-Arena`'s contamination percentage (2%) according to the exact match is significantly lower. When using the sentence similarity metric, we can effectively detect whether generated questions are just rephrases of existing questions. The percentage is largely reduced by 7% to 15% compared to other benchmarks.

## D   SYNTHETIC V.S. REAL-LIFE QUESTIONS

In this section, we try to show the generalizability of the synthetic questions in `Auto-Arena` to real-life questions.

**Design:**   The generated questions resemble real-world queries by design. In the question generation prompt, we specifically ask the examiner to draft questions that are "likely for a user to ask in real life". From Appendix B, we could also observe the similarity of the synthetic questions to real-life queries.

Table 8: Human Evaluation on Synthetic Questions and Real Questions.

|  | Volunteer 1 | Volunteer 2 |
|---|---|---|
| Correct | 27.1% | 38.9% |
| Incorrect | 27.1% | 11.9% |
| Cannot Tell | 45.8% | 49.2% |
| Agreement | -0.11 | |

**Human Study:**   To show that the generated queries are similar to real-life ones, we conduct the following human study. We compare 30 synthetic questions by `Auto-Arena` and 30 real-life questions. A human user is asked to look at a question randomly drawn and decide whether he/she believes that it is AI-generated, Real-Life, or if he/she cannot tell. The questions are collected in the Math category, where the 30 real-life ones are taken from MT-Bench (10 questions, drafted by experts), AMC-8 (4 problems, from the 2024 math competition), and AGI-Eval (16 math questions collected from college entrance exams). Two volunteers who are frequent users of LLMs and are familiar with AIGC participated. We report their respective results and agreement in Table 8. We can observe that humans cannot tell if the problems are synthetic almost half of the time. The user accuracy (correct percentages) is also low. We calculate the Cohen's Kappa agreement between the two users, which is -0.11. The agreement score shows that there is less agreement than random chance. The big divergence between human annotators' responses also shows subjectivity and uncertainty in the judgments. Therefore, we conclude that humans most likely cannot tell whether questions are synthetic or real-world, indicating small differences.

Table 9: Ablation Results on Synthetic Questions and Real Questions.

| Questions | GPT-4 Win Rate | Claude-3 Win Rate |
|---|---|---|
| Synthetic Questions | 80.00% | 20.00% |
| Real-life Questions | 75.86% | 24.14% |

**Ablation Study:** To validate the results' generalizability with real-world datasets, we conduct an ablation study comparing Auto-Arena's evaluation performances on real-life questions and synthetic questions. Specifically, we asked 2 candidates (GPT-4-Turbo-0409 and Claude-3-Haiku) to debate around 30 synthetic math questions and 30 real-world math questions (collected as in the human study shown in Table 8). If the results are generalizable, we would observe that the win rates of each model should be similar. The results are shown in Table 9. From the results, we can observe that the win rates of each model only differ by 4% on synthetic and real datasets, which shows consistent evaluation performances, validating the use of synthetic problems.

Aside from the supporting studies, the use of synthetic questions for evaluation has also been established as common practice. The Mathematics dataset (Hendrycks et al., 2021b) already uses synthetically generated math questions, where they note many advantages, such as the ease of providing a larger number of examples, the precise controls over difficulty levels, and the ease of testing generalization (since one can precisely vary different axes of difficulty in different question types). LMExamQA (Bai et al., 2024) also uses an LLM to generate questions in different domains. KI-Eval (Yu et al., 2024) asks an LM-powered interactor to generate questions. The list goes on. Using synthetic questions has become the common norm in NLP evaluation. Moreover, extensive experiments in `Auto-Arena` show high correlations with human results, which also demonstrates the alignment with real-world usage.

# E    ABLATION STUDY ON SELF-ENHANCEMENT BIAS OF THE QUESTION GENERATION STAGE

Table 10: Ablation Results on Self-Enhancement Bias for Question Generator.

| Questions | GPT-4 win rate | Haiku win rate |
|---|---|---|
| GPT-4 Generated Questions | 80.00% | 20.00% |
| Haiku Generated Questions | 76.92% | 23.08% |

We attempt to reduce self-enhancement bias of the question generation stage with explicit designs: Firstly, during question generation, we do not disclose to the examiner that it will participate in this tournament and we do not ask the examiner to generate only questions that can be solved by itself. Secondly, the peer-debate process further reduces bias in initial question generation: Debating ensures that candidates are evaluated not only on their response to the initial question, but also in more comprehensive and deeper abilities, such as strategizing, criticizing the opponent, and drafting questions. In other words, answering the initial question well does not necessarily win a whole debate. In the debate design in Figure 2, candidates also have a "raise" action, where they ask questions to the opponent. This process essentially decentralizes the question-generation process.

To systematically examine whether self-enhancement bias is present. We conduct an ablation study: We examine enhancement bias with 2 models as an example: GPT-4 (GPT-4-turbo) and Haiku (Claude-3-Haiku). Firstly, we ask GPT-4 and Haiku to generate 30 math questions separately. Then, we conduct peer debates between the two candidates (GPT-4 and Haiku) on both sets of questions and evaluate results with the best-5-LLM committee as in the main experiments.

We evaluate the performance differences from the evaluation results: If self-enhancement bias is low, the ranking achieved should remain the same. In other words, the weaker model will always lose, even on the questions generated by itself.

The ablation results are shown in Table 10. From the results, we can observe that, in both sets of generated questions, the GPT-4 win rate remains significantly higher than the Claude-3-Haiku

win rate. Even if some limited extent of self-enhancement bias is present, the result difference is significant enough to reach the correct ranking.

## F  PER-CATEGORY ANALYSIS ON SPECIFIC MODEL'S PERFORMANCES

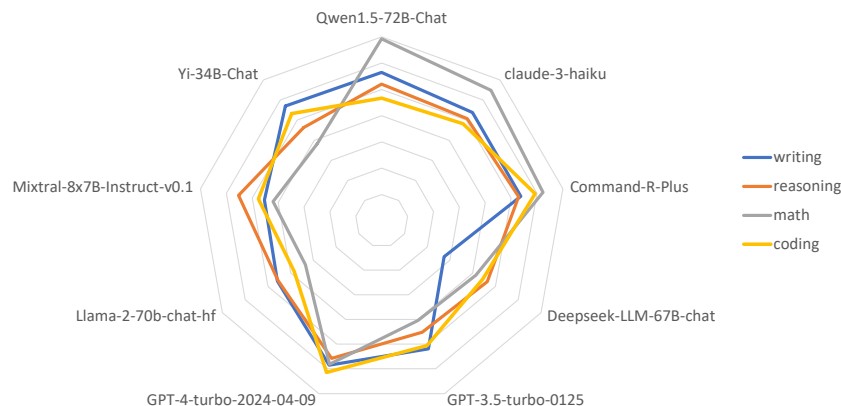

Figure 10: ELO Scores across Different Models on 4 Representative Categories.

`Auto-Arena` could be used to estimate performances in different domains. As an example, we provide an analysis of model performances across four representative domains in Figure 10. Out of the 8 domains in the main experiment shown in 3.2, we plot the four domains in which the model performances diverge the most from overall scores into a radar chart. In the math domain, `Auto-Arena` evaluates Qwen-1.5 to have a stronger edge compared to other models. However, Qwen-1.5 also shows degrading performances in other domains, such as coding. GPT-4-Turbo, on the other hand, shows equally strong performances in all domains. While Deepseek-LLM-67B shows average performance for most tasks, it lags behind in the writing domain, which degrades its overall performance.

## G  INTER-JUDGE AGREEMENT

As shown in Figure 11, the Cohen's Kappa agreement (McHugh, 2012) among judges before committee discussion is very low, averaging 0.16, which indicates slight agreement. We notice that

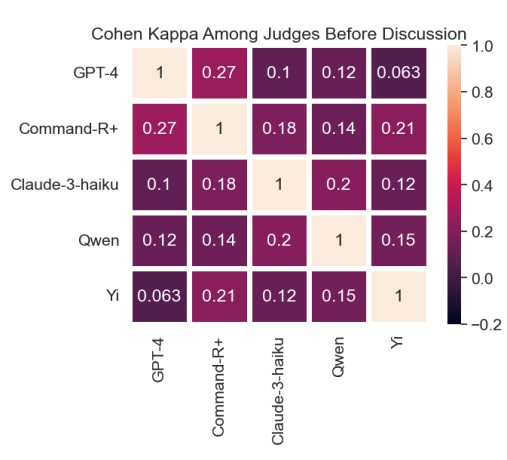

Figure 11: Cohen's Kappa Agreement with Majority Vote Before Committee Discussions.

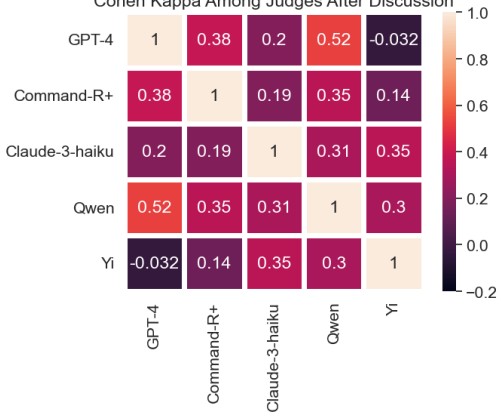

Figure 12: Cohen's Kappa Agreement with Majority Vote After 1 Round of Committee Discussion.

weak model judges and strong model judges has an especially low agreement, such as GPT-4 and Yi. This shows that general model capabilities could result in significant performance gaps when used as judges.

After the 1 round of communication, agreements significantly improved as the judges become convinced by more persuasive arguments. The average Cohen's Kappa after discussion reaches 0.27, which indicates fair agreement.

## H    MODEL SELECTION FOR THE MAIN EXPERIMENT

Table 11: Model Selection for the Main Experiment. "Newest" and "Strongest" refer to the state at the time of experiments (2024 April). Bolded models are selected for the primary experiment with 7 models. Unbolded models are the ones added during extension.

| Model Name | Reasons for Inclusion | License |
|---|---|---|
| **GPT-4-0409-Turbo** (OpenAI et al., 2024) | Newest and Strongest in GPT model family under GPT-4 | Proprietary |
| GPT-4o-2024-05-13 (Openai, 2024b) | Newly released model in GPT Model Family | Proprietary |
| **GPT-3.5-Turbo-0125** (Openai, 2024a) | Newest ChatGPT version in the GPT Model Family | Proprietary |
| Claude-3.5-Sonnet-20240620 (Anthropic, 2024) | Newest in Claude model family under Claude-3.5 | Proprietary |
| **Claude-3-Haiku** (Anthropic, 2024) | Newest and Cheapest in Claude model family under Claude-3 | Proprietary |
| Qwen/Qwen2-72B-Instruct (Bai et al., 2023) | Representative of Qwen Model Family under Qwen-2 | Proprietary |
| **Qwen1.5-72B-chat** (Bai et al., 2023) | Representative of Qwen model family under Qwen-1.5 | Qianwen LICENSE |
| **Command R Plus** (Cohere, 2024) | Strongest model in Command R Model Family | CC-BY-NC-4.0 |
| Llama-3-70b-chat-hf (Meta, 2024) | Representative of Llama Model Family under Llama-3 | Llama 3 Community |
| **Llama-2-70b-chat** (Touvron et al., 2023) | Representative of Llama Model Family under Llama-2 | Llama 2 Community |
| **Mixtral-8x7b-Instruct-v0.1** (Jiang et al., 2024) | Strongest in open-source Mistral small models MOE Structure | Apache 2.0 |
| Gemma-2-27b-it (Team et al., 2024a) | Representative of the Gemma family | Apache 2.0 |
| Gemini-1.5-flash-exp-0827 (Team et al., 2024a) | Cheapest in the Gemini-1.5 family | Proprietary |
| **Yi-34B-Chat** (AI et al., 2024) | Strongest in Yi Model Family on Chatbot Arena | Yi License |
| **Deepseek-LLM-67B-chat** (DeepSeek-AI et al., 2024) | Representative open-source model in Deepseek Family | DeepSeek License |

In Table 11, we show all the models selected for the main experiment and expansion. We also include the reasons for selection. Overall, we try to select a representative set of famous models on Chatbot Arena top 20 list. While the Chatbot Arena ranking mostly consists of models with different versions, we only select the strongest or newest model from each model family. Besides the models on Chatbot Arena, we include 4 under-evaluated famous Chinese models to investigate their performances.

## I    COMPARISON AND COSTS OF BASELINE METHODS AND AUTO-ARENA

Table 12 shows a comparison between benchmark evaluation methods and `Auto-Arena`. Compared to previous methods, the main advantage of `Auto-Arena` is the zero need for human dataset construction or intervention and the freshness of queries. Another innovation compared to previous model-based systematic benchmarking procedures is using a committee of LLMs to discuss and vote for a final winner, which introduces diverse viewpoints. The most important innovation of `Auto-Arena` is the peer-battle mechanism, which asks LLM agents to compete and debate with each other. The resulting evaluation on the multi-turn debate then becomes more in-depth, interactive, and comprehensive.

For the evaluation cost, the costs of `Auto-Arena` are on the same scale as other benchmarks: We note that the primary experiment among 9 models costs around $45 USD. Therefore, the estimated cost is $5 per model. As models on the ranking board increase, the costs of conducting debates should grow slowly in log scale, which comes from conducting $nlog_2(n)$ pairings when adding 1 model to a ranking of $(n-1)$ models. The evaluation costs, however, shall remain the same as we use a committee of 5 LLMs at all times.

To help better understand the computational cost breakdown for each component, we estimate the computational resources for each component based on input/output tokens in Table 13. For example, if all agents (candidates and judges) have costs and inference times that are on par with GPT-4o, the API costs would be USD 0.22 per evaluation question. Evaluating our set of 40 questions would cost USD 8.8. In the tournament, however, cheaper and non-proprietary models are engaged as well, which drives down the costs.

Table 12: Comparison between `Auto-Arena` and Other Benchmarks.

| Method | Manual Construction of Queries | Freshness | Eval. Cost per Model | Judge |
|---|---|---|---|---|
| OpenLLM Leaderboard (Beeching et al., 2023) | Yes | Static | - | Answer Accuracy |
| MMLU (Hendrycks et al., 2021a) | Yes | Static | - | Answer Accuracy |
| GPQA (Rein et al., 2023) | Yes | Static | - | Answer Accuracy |
| LC-AlpacaEval (Dubois et al., 2024a) | Yes | Static | $10 | Single LLM (GPT-4) |
| MT-Bench (Zheng et al., 2023) | Yes | Static | $10 | Single LLM (GPT-4) |
| Arena Hard (Li* et al., 2024) | Yes | Frequent Updates | $25 | Single LLM (GPT-4) |
| Chatbot Arena (Zheng et al., 2023) | Yes | Live | Very High | Humans |
| **Auto-Arena** | **No** | **Freshly Generated** | **$5** | **Committee of LLMs** |

Table 13: Computational Cost Breakdown for Each Component in `Auto-Arena` Framework

| Role | Step | Input tokens (Avg) | Output tokens (Avg) |
|---|---|---|---|
| Examiner | Question Generation | 25 | 38 |
| Candidate | Peer Debate | 7778*2 candidates | 1330*2 candidates |
| Judges | Round 1 Verdicts | 5224*5 judges | 178*5 judges |
| | Round 2 Verdicts | 5937*5 judges | 142*5 judges |
| Total | | 71386 | 4298 |

Analyzing the table carefully, we see that question generation is the cheapest component. The biggest effort is actually on the committee judgments and discussions, where bringing in several judges to discuss increases the costs.

