# OpenReview forum: "Auto-Arena: Automating LLM Evaluations with Agent Peer Battles and Committee Discussions"
_ICLR.cc/2025/Conference — Submitted to ICLR 2025_

### Official Review · Reviewer_j7qN · 2024-10-23

**Soundness:** 2
**Presentation:** 3
**Contribution:** 2
**Rating:** 5
**Confidence:** 4

**Summary:**

The paper proposes Auto-Arena, an evaluation framework that automates the evaluation of large language models (LLMs) through a multi-stage process involving question generation, peer battles, and committee discussions. This approach mimics human-like evaluation while addressing limitations of static datasets and manual assessments. Auto-Arena achieves high correlation with human preferences, enhancing the reliability of LLM evaluation. It also proves adaptable to different languages and domains, making it suitable for diverse LLMs and scalable leaderboard creation.

**Strengths:**

- Auto-Arena provides a scalable, automated evaluation of LLMs, reducing reliance on manual efforts.
- Works across different languages and tasks, making it versatile.
- By using the peer-battle mechanism, the framework offers a competitive context that reveals subtle model strengths and weaknesses.

**Weaknesses:**

- The code provided for review has not been anonymized properly and reveals the author's name: https://anonymous.4open.science/r/Auto-Arena-Code/analysis_scripts/result_analysis.ipynb
- The paper only involves baselines experiments for automated LLM evaluations using the Auto-Arena framework. While the study introduces a modified evaluation method, the results primarily showcase this framework's effectiveness compared to existing evaluation methods, rather than demonstrating significant new findings about model capabilities, which makes the contributions seem incremental. For example, it would be nice to see the individual model performances on each domain according to Auto-Arena, and some analysis on it.
- It seems like Auto-Arena involves a lot more agents interacting (used in both peer debate and committee discussions), making it more resource-intensive than existing evaluation methods. Does the increase in accuracy justify the added computational costs? How does Auto-Arena compare in cost-efficiency to existing evaluation methods? It would be nice to see a more detailed cost breakdown and comparison, including computational resources used, to better assess the cost-efficiency trade-off between Auto-Arena and existing evaluation methods.

**Questions:**

See above weaknesses.

---

> ### Author Response · Authors · 2024-11-19
> **Thank you for the comments. We reply to some of the concerns here.**
>
> We thank the reviewer for the comments and would like to reply to some of the concerns below:
>
> 1. **Code anonymity.**
>
> Thank you for pointing it out. Upon closer inspection, The warnings displayed in one of the jupyter notebook cells may risk revealing the identity of an author. There was no critical information such as full name or institution present. We apologize for the oversight and have fixed it.
>
> 2. **The paper only shows “this framework's effectiveness compared to existing evaluation methods”, which seems incremental. Better to show “significant new findings about model capabilities”.**
>
> First of all, the primary contribution of the paper is Auto-Arena, an automated evaluation framework. It proposes a method to tackle the **task of LLM Evaluation**.
>
> Therefore, task performance is measured by the evaluation method’s effectiveness compared to previous ones. Compared to previous evaluation methods, Auto-Arena is the first to fully automate the LLM evaluation process in an end-to-end way. As no human efforts is involved, it still reaches the highest correlation with human preferences, validating the evaluation method’s robust performance. In the age of fast-developing LLMs, the fully automated feature opens up possibilities for deriving fast evaluations without manual efforts.
>
> As to your suggestion of showing “significant new findings about model capabilities”, we would like to note that the results of the evaluation (specific model capabilities, etc) are not the focus of the paper. The focus is the **evaluation method** itself. Therefore, the results are presented and analyzed, but not studied in detail.
>
> 3. **“Cost efficiency compared to previous methods”? “Detailed breakdown”?**
>
> A cost analysis is already provided in Appendix H of the paper. As shown in Table 12, Auto-Arena’s evaluation costs around USD 5 for one model, which is lower than previous LLM-judge-based evaluation methods. Despite the lower costs, the evaluation performance (correlation with human preferences) is higher.
>
> As to your question on the detailed computational resource breakdown, during our tournament, many different models are used and they all have various API costs and inference times. The $5 was derived from the resulting overall cost for our main experiment.
>
> To help better understand the computational resources, we have also summarized the table below, which estimates the computational resources for each component based on input/output tokens. Please see our summarized table below:
>
> | Role  | Step | Input tokens (Avg) | Output tokens (Avg) |
> | -------- | ------- | ------- | ------- |
> | Examiner  | Question Generation | 25 | 38 |
> | Candidate | Peer Debate | 7778*2 candidates | 1330*2 candidates |
> | Judges | Round 1 Verdicts | 5224*5 judges | 178*5 judges |
> | | Round 2 Verdicts | 5937*5 judges | 142*5 judges |
> | Total | | 71386 | 4298 |
>
> Table 1. Cost breakdown (in terms of input and output tokens) for one evaluation (a comparison between two models on one question)
>
> For example, if all agents (candidates and judges) have costs and inference times that are on par with GPT-4o, the API costs would be USD 0.22 per evaluation question. Evaluating our set of 40 questions would cost USD 8.8. In the tournament, however, cheaper and non-proprietary models are engaged as well, which drives down the costs.
>
> Analyzing the table carefully, we see that question generation is the cheapest component. The biggest effort is actually on the committee judgments and discussions, where bringing in several judges to discuss increases the costs.
>
> We thank the reviewer again for his/her comments. If there are remaining concerns or lingering questions, we’d be happy to discuss them in more detail.

---

> > ### Comment · Reviewer_j7qN · 2024-11-25
> > **Response to authors' rebuttal**
> >
> > > we would like to note that the results of the evaluation (specific model capabilities, etc) are not the focus of the paper. The focus is the evaluation method itself.
> >
> > A major goal of building evaluation benchmarks is to inform users about the specific strengths and weaknesses of models, helping them make informed decisions based on their specific needs. While Figures 4, 5 and 6 provide Elo ratings and rankings of the models, they only give a high-level view of relative performance. More detailed insights or even speculations about model behavior across different domains (besides only reiterating the experimental results shown in the figures) will be appreciated.
> >
> > How might the experiment results help users understand model-specific performance insights beyond the rankings?

---

> ### Author Response · Authors · 2024-11-23
> **May we check whether we have addressed the questions to your satisfaction?**
>
> Dear reviewer j7qN,
>
> Thank you once again for taking the time to review our paper. Could you let us know whether the questions have been addressed to your satisfaction? If not, we'd be happy to discuss further! If it has, would you mind adjusting your current ratings?
>
> Best regards,
> Authors

---

> ### Author Response · Authors · 2024-11-25
> **Reply**
>
> Thank you for the reply, we reply to your concerns here:
>
> Q. **goal of building evaluation benchmarks is to inform users about the specific strengths and weaknesses of models**.
>
> R: We'd like to respectively clarify that our paper proposes a comprehensive framework that can be used to evaluate LLM capabilities and performances. We verify Auto-Arena's effectiveness by measuring the alignment with human preferences, where we show consistent improvements in correlation metrics. **The method can be expanded to various domains and cover many LLMs**. **Previous important LLM evaluation papers also focus on the evaluation method itself, without analyzing specific model results**: G-Eval[1] conducts experiments and compares to correlations to human judgments of other meta-evaluation benchmarks. MT-Bench[2] compares the LLM judge's performances across different judge designs and evaluates the agreement with humans. It does not talk about pitfalls and strengths of specific models. AlpacaEval[3] shows correlations with Chatbot Arena compared to other benchmarks.  ChatEval[4] shows correlations and accuracy of the evaluation results compared to other benchmarks. The list goes on. Therefore, we believe it's unfair to reject the paper on the grounds that it doesn't analyze the weaknesses and strengths of specific models.
>
> Moreover, besides quantitative evaluation, we do analyze the framework and the behaviors of specific debates and models. As mentioned in L403-405, we notice that models claimed to have multi-lingual efficiency perform better on the Chinese experiment compared to the English one. In section 5, we investigate various behaviors of the LLM candidates and highlight specific strengths of the models.
>
> Q. **How might the experiment results help users understand model-specific performance insights beyond the rankings?**
>
> R. In the main experiment, our goal is to derive a leaderboard on general LLM capabilities, which is similar to [Chatbot Arena](https://lmarena.ai/?leaderboard). The main advantage is that it's fully automated and, therefore, can derive timely evaluations of different LLMs. If, for example, besides a quantitative metric, the user would like to understand specific behaviors of specific models during debates, we release the debate samples online (link included in paper footnote 3), which the users can use to derive subjective observations on further model improvement. If the users are interested in deriving the leaderboard for model performance in a specific domain, Auto-Arena can be extended to other domains or languages by altering the language or aspect specification of the prompts. We provided Chinese as an example in the main paper section 4.2.
>
> [1] G-EVAL: NLG Evaluation using GPT-4 with Better Human Alignment
>
> [2] Judging LLM-as-a-Judge with MT-Bench and Chatbot Arena
>
> [3] Length-Controlled AlpacaEval: A Simple Way to Debias Automatic Evaluators
>
> [4] CHATEVAL: TOWARDS BETTER LLM-BASED EVALUATORS THROUGH MULTI-AGENT DEBATE

---

> > ### Author Response · Authors · 2024-11-25
> > **[URGENT] your engagement is needed!**
> >
> > Dear Reviewer j7qN,
> >
> > We hope this message finds you well. The discussion period is ending soon, I am writing to emphasize the importance of your review for our submission. Your initial score is significantly lower than the other three reviewers, and we believe this discrepancy may indicate a misunderstanding or oversight.
> >
> > We have addressed all the concerns in our detailed rebuttal and the further clarifications. We would appreciate your prompt attention to it. A thorough reassessment is crucial to ensure a fair evaluation.
> >
> > Your expertise is highly valued, and we trust that a reconsidered review will reflect the true merit of our work.
> >
> > Thank you for your immediate attention to this matter.
> >
> > Best regards, Authors

---

> ### Comment · Reviewer_j7qN · 2024-11-26
> **Response to Authors**
>
> > Previous important LLM evaluation papers also focus on the evaluation method itself, without analyzing specific model results
>
> In Table 7 of MT-Bench [1], they gave a clear breakdown of the win rates of each model on different categories (roleplay, math, coding, etc.) In their analysis, they said "we see GPT-4 is significantly better than others. Vicuna-13B is noticeably worse than GPT-3.5/4 in reasoning, math, and coding categories. Note that in math/coding category, GPT-3.5 and GPT-4 have similar overall
> win-rate because they both failed to answer some hard questions, but GPT-4 is still significantly better than GPT-3 in the direct pairwise comparison or single-answer grading." This is informative in telling users the specific model results that I was referring to.
>
> It seems like the prompts used for experiments in Auto-Arena are broken down into different domains/categories, similar to what MT-Bench did. However, there are no or very little analysis on how the models perform on these different domains. Case studies would've helped in explaining this part as well.
>
> [1] Judging LLM-as-a-Judge with MT-Bench and Chatbot Arena

---

> > ### Author Response · Authors · 2024-11-26
> > **We have revised our paper to reflect your suggestions**
> >
> > Dear reviewer j7qN,
> >
> > Thank you for the reference. We recognize that some readers may hope to refer to an analysis of the results. We have included a radar chart of model performances in 4 representative categories (writing, reasoning, math, and coding) along with a detailed analysis in Appendix F of the newly revised submission.
> >
> > Moreover, pertaining your earlier comments, we have also included the aforementioned cost breakdown of different framework components in Appendix I.
> >
> > Please let us know whether we have addressed your concerns to your satisfaction. If not, we are happy to discuss further.
> >
> > Best regards,
> >
> > Authors

---

> > > ### Comment · Reviewer_j7qN · 2024-11-26
> > > **Response to Authors**
> > >
> > > Thanks for the additional analysis on model performances and cost breakdown. There seems to be a typo in Appendix F (GPT-5 -> GPT-4-Turbo).
> > >
> > > However, the authors have addressed my concerns, thus I have raised my scores accordingly.

---

> > > > ### Author Response · Authors · 2024-11-27
> > > > **Thank you for the recognition, any additional feedback?**
> > > >
> > > > Dear Reviewer j7qN,
> > > >
> > > > Thank you for pointing out our typo, we have revised the typo in the newly uploaded rebuttal revision. As the deadline to update paper revisions approaches, would you mind letting us know about your remaining questions or lingering concerns?
> > > >
> > > > As other reviewers gave positive scores, we would love to understand your concerns better for improving our paper and try our best to address them in the next few days.
> > > >
> > > > Sincerely,
> > > >
> > > > Authors

---

### Official Review · Reviewer_PycX · 2024-10-28

**Soundness:** 3
**Presentation:** 3
**Contribution:** 2
**Rating:** 6
**Confidence:** 4

**Summary:**

Auto-Arena uses LLM-driven agents to handle the entire evaluation process through debates and discussions, cutting down the need for manually labeled data. This framework closely aligns with human preferences and is flexible enough to adapt across different domains and languages, showing robust performance in both English and Chinese evaluations. The process has three main stages:
1. An LLM examiner dynamically creates varied questions to prevent data contamination and simulate real-world use cases.
2. Two LLM candidates debate across multiple rounds on each question, showcasing their abilities in reasoning, interaction, and strategy.
3. A panel of top-performing LLMs jointly assesses the debates to determine the winner, enhancing fairness and reducing biases in the evaluation process.

**Strengths:**

1. The paper provides solid experimental validation for Auto-Arena, showing it aligns with human preferences at a correlation of 92.14%. In terms of experimental design, the authors included multiple benchmarks and ablation studies, which effectively demonstrate how multi-round battles and committee discussions enhance evaluation accuracy and consistency.

2. In the appendix, the authors mention that their method is cheaper than comparable approaches (though still more expensive than dataset-based evaluation), estimating around $5 to evaluate a single model. However, they don't break down this calculation, which seems like a pretty relevant detail. Could you elaborate on that? I think it would really strengthen the paper to clarify the cost calculation.

**Weaknesses:**

1. Using LLMs to evaluate other LLMs has always been considered somewhat unreliable, as their effectiveness as judges is closely tied to the foundational model’s performance. In classic tasks where these models perform well as judges, it’s often due to data leakage (i.e., the model has likely seen the answers before). The authors claim that Auto-Arena can be “easily extended to other domains and languages,” but there’s a lack of experimental support for this. Testing on newer tasks and data—ones that no LLM agent has seen—would better demonstrate if it can still perform well in unfamiliar contexts.

2. Although Auto-Arena aligns closely with human preferences, the paper could improve by analyzing instances where LLM judgments differ from human judges, especially in nuanced or high-stakes cases. Examples or further analysis in these areas could highlight where LLM judges may still need refinement, helping the framework align more closely with human evaluation standards.

**Questions:**

1. As I mentioned earlier, the paper states that Auto-Arena is cheaper than other methods but still costs more than traditional dataset-based evaluations, estimating around $5 per model evaluation. Could you provide a more detailed breakdown of this cost? Specifically, how much goes to computational resources, model inference, multi-round battles, and committee discussions? In addition to the computational cost, what about the time required for these evaluations?

2. Why did you choose stronger models as judges instead of the initial participants in the debate? Wouldn’t using the original debaters as judges improve the quality of candidate answers?

3. Auto-Arena is presented as a dynamic, automated alternative to static benchmarks, but it still relies heavily on the strength of the foundational model itself. Have you considered incorporating external information sources, similar to how Bing does, to enhance performance?

---

> ### Author Response · Authors · 2024-11-19
> **Thank you for the comments. We reply to some of the concerns here. (1/2)**
>
> We thank the reviewer for the constructive comments and would like to reply to some of the concerns below:
>
> **Weakness 1: Evaluations could be “unreliable”,  “tied to the foundational model’s performance”.**
>
> We agree with your point: the performance of a single judge model may be inconsistent. This is exactly the motivation behind our multi-judge design: We incorporate multiple strong-performing judges to engage in discussions to improve judgment performance. As shown in ablation (“w/o committee discussion” in Table 2), committee discussion improves judgment correlations with human preferences.
>
> **“Data Leakage”.**
>
> We also agree with the data contamination concerns you have raised. Therefore, by dynamically drafting and updating the questions, Auto-Arena is more contamination-proof compared to static benchmarks, such as MMLU. We show this effect in the contamination analysis in Appendix C.
>
> **“Easily extended to other domains and languages”?**
>
> To clarify this claim, Auto-Arena would be quite useful for extending into many of the fine-grained usage scenarios currently. For example, if a gaming company wants to use LLMs for NPC dialogue, this would be a very fine-grained application where no datasets have been developed before. In this case, Auto-Arena can help develop a reliable evaluation. In this sense, we believe Auto-Arena would be immensely helpful to the industry that tries to incorporate LLMs.
>
> Due to the limited resources, we only included Chinese as an example (Section 4.2) for the extension.
>
> As for your comment on “testing on newer tasks and data—ones that no LLM agent has seen”, we are afraid this would not be feasible. In order for the agents to function properly (engage in debates, draft the questions, etc), the agents have to grasp a certain degree of prior knowledge of the task. For example, the agents can evaluate “Chinese performance” because it has seen and known Chinese during pre-training. If it has never seen this language before, it would not be able to “speak” Chinese, let alone debate in Chinese.
>
> **Weakness 2: “Analyzing instances where LLM judgments differ from human judges”.**
>
> Thank you for this note. There are papers that have specifically studied how LLM-based judges differ from humans. [1] discovers that LLM judges possess Misinformation Oversight Bias, Gender Bias, Authority Bias, and Beauty Bias. In contrast, humans also have strong Misinformation Bias and Beauty Bias, but very minimal Gender Bias.
> As Auto-Arena is also built with LLM-as-a-judge, the differences with human judgments would be quite similar. Although this is not the main focus of the paper, we agree that such case studies would make for an interesting case study and would aim to incorporate such a study on the website or in a later version of the paper.
>
>
> **Question 1: Detailed Computational Resources?**
>
> Thank you for your question about the detailed breakdown of costs. During our tournament, many different models are used and they all have various API costs and inference times. The $5 was derived from the resulting overall cost for our main experiment.
>
> To help better understand the computational resources, we have also summarized the table below, which estimates the computational resources for each component based on input/output tokens. Please see our summarized table below:
>
> | Role  | Step | Input tokens (Avg) | Output tokens (Avg) |
> | -------- | ------- | ------- | ------- |
> | Examiner  | Question Generation | 25 | 38 |
> | Candidate | Peer Debate | 7778*2 candidates | 1330*2 candidates |
> | Judges | Round 1 Verdicts | 5224*5 judges | 178*5 judges |
> | | Round 2 Verdicts | 5937*5 judges | 142*5 judges |
> | Total | | 71386 | 4298 |
>
> Table 1. Cost breakdown (in terms of input and output tokens) for one evaluation (a comparison between two models on one question)
>
> For example, if all agents (candidates and judges) have costs and inference times that are on par with GPT-4o, the API costs would be USD 0.22 per evaluation question. Evaluating our set of 40 questions would cost USD 8.8. In the tournament, however, cheaper and non-proprietary models are engaged as well, which drives down the costs.
>
> Analyzing the table carefully, we see that question generation is the cheapest component. The biggest effort is actually on the committee judgments and discussions, where bringing in several judges to discuss increases the costs.

---

> ### Author Response · Authors · 2024-11-19
> **Thank you for the comments. We reply to some of the concerns here. (2/2)**
>
> **Question 2: Why not “using the original debaters as judges”?**
>
> We explicitly avoid such a scenario to reduce self-enhancement bias [2], which is where the LLM judges would prefer the answers generated by themselves. Therefore, to select judges, we follow the 2 criteria: 1) the judge is not from the same model family as the current debaters (to avoid self-enhancement bias. 2) at each run, we select the 5 top-performing models from the current leaderboard. Note that the current leaderboard consists of all debaters that have participated so far. We select the better-performing as the better LLMs could serve as better judges (shown in Appendix F).
>
> **Question 3: “Incorporating external information sources to enhance performance?”**
>
> Thank you for the suggestion. It would be a good direction to consider in the future when we are evaluating information-rich questions that require retrieval from external sources. Then, introducing RAG could aid in the LLM judge’s ability to diagnose factual mistakes. As most of the current tasks we evaluate (math, coding, writing, roleplay, etc) are not very reliant on external knowledge, we haven’t included such mechanisms so far. But we would be happy to explore such directions in the future.
>
> Thank you again for the helpful comments. Please let us know if the above has clarified some of your concerns or if there are still lingering concerns. We would be happy to address them.
>
>
> [1] Humans or LLMs as the Judge? A Study on Judgement Biases.
>
> [2] Justice or Prejudice? Quantifying Biases in LLM-as-a-Judge.

---

> ### Author Response · Authors · 2024-11-23
> **May we check whether we have addressed the questions to your satisfaction?**
>
> Dear reviewer PycX,
>
> Thank you once again for taking the time to review our paper. Could you let us know whether the questions have been addressed to your satisfaction? If not, we'd be happy to discuss further! If it has, would you mind adjusting your current ratings?
>
> Best regards,
> Authors

---

> > ### Comment · Reviewer_PycX · 2024-11-23
> > **Comment**
> >
> > I appreciate the explanations, but some of the answers are not answering what I have asked:
> >
> > __R1__ "As for your comment on “testing on newer tasks and data—ones that no LLM agent has seen”, we are afraid this would not be feasible. In order for the agents to function properly (engage in debates, draft the questions, etc), the agents have to grasp a certain degree of prior knowledge of the task. For example, the agents can evaluate “Chinese performance” because it has seen and known Chinese during pre-training. If it has never seen this language before, it would not be able to “speak” Chinese, let alone debate in Chinese."
> >
> > __Q1.__ I mean how might Auto-Arena handle tasks that require highly specialized or emerging knowledge (e.g., newly reported events)? Basically, if a task only requires common sense, we would not really need LLMs to do the discussion. In real-world application scenarios, we often need to work on something requires highly specialized or emerging knowledge. How does Auto-Arena perform in those cases?
> >
> >
> > __R2__ "Thank you for this note. There are papers that have specifically studied how LLM-based judges differ from humans. [1] discovers that LLM judges possess Misinformation Oversight Bias, Gender Bias, Authority Bias, and Beauty Bias. In contrast, humans also have strong Misinformation Bias and Beauty Bias, but very minimal Gender Bias. As Auto-Arena is also built with LLM-as-a-judge, the differences with human judgments would be quite similar. Although this is not the main focus of the paper, we agree that such case studies would make for an interesting case study and would aim to incorporate such a study on the website or in a later version of the paper."
> >
> > __Q2__ LLMs are found to be good generators but sometimes not effective as verifiers. The authors avoid discussing when their model might not align with human judges. It is acceptable to acknowledge these limitations so that future researchers can fully understand the implementation and its potential challenges.

---

> ### Author Response · Authors · 2024-11-25
> **Reply**
>
> Thank you for the reply and concerns. We really appreciate the opportunity to discuss.
>
> Q1. I mean how might Auto-Arena handle tasks that require highly specialized or emerging knowledge (e.g., newly reported events)? Basically, if a task only requires common sense, we would not really need LLMs to do the discussion. In real-world application scenarios, we often need to work on something requires highly specialized or emerging knowledge. How does Auto-Arena perform in those cases?
>
> R1. **Currently, Auto-Arena focuses on testing the general capabilities of LLMs**, i.e. how helpful it is to human users. The tasks span domains such as writing, roleplay, and math.
>
> If we are to extend to domains that require highly specialized knowledge, we believe it would then be beneficial to enable RAG for all the agents, which could incorporate external knowledge into this framework. RAG would allow the debaters to utilize specialized knowledge and examiners to detect discrepancies accurately. This would be an interesting direction to explore in the future.
>
> Even besides the RAG component, the judges can use other dialogue techniques to evaluate the quality of the debate when they don't possess the specialized knowledge themselves. For example, in paper [1], the examiner can interact with the debaters and employ a cross-examination technique to detect the candidate's loopholes in logic and memory.
>
> Therefore, it is a possible and interesting use case to extend into scenarios that require specialized knowledge. But to ensure performance, there would be other designs and techniques involved to guarantee the performance of the judges. We will consider such extensions in the future.
>
>
> Q2 LLMs are found to be good generators but sometimes not effective as verifiers. The authors avoid discussing when their model might not align with human judges. It is acceptable to acknowledge these limitations so that future researchers can fully understand the implementation and its potential challenges.
>
> We will include the previously mentioned biases caused by LLM-as-a-judge **in the limitations section** as possible pitfalls of Auto-Arena. Thank you for raising it.

---

> > ### Author Response · Authors · 2024-11-25
> >
> > Dear reviewer PycX,
> >
> > I hope this message finds you well. The discussion period is ending soon, I am writing to emphasize the importance of your review for our submission.
> >
> > We have addressed all the concerns in our detailed rebuttal and the further clarifications. We would appreciate your prompt attention to it. A thorough reassessment is crucial to ensure a fair evaluation.
> >
> > Your expertise is highly valued, and we trust that a reconsidered review will reflect the true merit of our work.
> >
> > Thank you for your immediate attention to this matter.
> >
> > Best regards, Authors

---

> > ### Author Response · Authors · 2024-11-26
> > **We have revised our paper according to your valuable suggestions.**
> >
> > Dear reviewer PycX,
> >
> > We have uploaded a new rebuttal revision to reflect your suggestions in the following aspects:
> >
> > 1. Corresponding to your concern on the Detailed Computational Resources, we have included the rebuttal results in Appendix I. It now includes a table of detailed breakdown costs for each framework component in Auto-Arena and an analysis on the costs.
> >
> > 2. Corresponding to your input on LLM judge models that might not align with human judges, we included it as a limitations point in L538-539 to raise readers' awareness on this point.
> >
> > Please let us know whether we have address the concerns to your satisfaction. We look forward to hearing from you.
> >
> > Best regards,
> >
> > Authors

---

> > > ### Comment · Reviewer_PycX · 2024-11-26
> > >
> > > Thank the authors for their responses. I have increased my score to 6 for the effort, but I feel it is more like a 5.5—really borderline work.

---

### Official Review · Reviewer_MUxT · 2024-11-02

**Soundness:** 4
**Presentation:** 4
**Contribution:** 3
**Rating:** 6
**Confidence:** 3

**Summary:**

The paper proposes a new framework name "Auto-Arena", the core idea is make evaluations dynamic and rely fully on static datasets for evaluating LLMs as new models and datasets gets introduced. The authors make use of different set models in agent framework to make the models act like peers in a debate and then use LLM as jusge to judge the conversation or the debate the previously occurred between the models. The results are good when compared to the existing baselines and detailed analysis is done in terms of contaminations, ablation study etc.

**Strengths:**

-> The architecture of the new framework is novel

-> Using questioning as part of the framework is something new and would help in coming works, as questioning is an important aspect in terms of reasoning

-> The results are well above the existing baseline results

-> Detailed analysis of contamination results

-> Evaluation is very big problem in the LLM research field and evaluations like this help in making it easier for evaluating LLMs as more
and more models are released

-> The paper is very well written

**Weaknesses:**

-> The Limitations section is missing, while the work is good there are limitation of this work as well

-> The results would have been more robust if there was human in loop evaluations rather than comparing with static baselines

-> The evaluation for the questions is limited to only 30 questions, there should have more questions taken into consideration for the evaluation of the question as there questions are the most important part for this framework, they lay the foundation. Also more detailed analysis of these questions in the main paper would be better, given the importance of this part.

-> There is need for exploring the models for unseen data and how the models can debate or judge on the topic they have not seen before, you would need all the models to have equal time-stamp cutoffs to consider for peer-review or judge the evaluations

**Questions:**

What would happen if you use one weak model or a small model in the framework? how would the results change if that model is in peer-battle and when it is acting as judge and when it is creating questions?

---

> ### Author Response · Authors · 2024-11-23
> **Thank you for the review. Please see our responses below.**
>
> We thank the reviewer for his/her constructive comments and would like to address some of the concerns below.
>
> 1. **Limitations?**
>
> We didn’t include limitations discussion as it wasn’t required by ICLR. However, we have included limitations in an earlier version and would be glad to add it back to the discussion in the conclusion section:
>
> “There still exist some limitations to the current approach. For example, the distribution of question domains is artificially designed, which may deviate from real-life distributions. Currently, Auto-Arena focuses on 1-to-1 peer battles, which limits its usage in multi-player scenarios.”
>
> 2. **“Human in loop evaluations rather than comparing with static baselines”**
>
> Do you mind clarifying what the question refers to? If you mean that we should consider comparing to human-based evaluations, we’d like to note that the main results of Auto-Arena is derived exactly based on its correlation to human preference ratings, where we take chatbot arena as ground truth. Besides static baselines, we also compare to model-based ones (Alpacaeval, MT-Bench, Arena-Hard).
>
> If you mean that we should include human in the loop in Auto-Arena, it would conflict with our overall motivation of eliminating human efforts in the task of LLM evaluation.
>
> 3. **“30 questions” only? More detailed analysis of these questions?**
>
> As noted in the L258-267, we include 40 questions, with 5 from each of the 8 categories. This number was selected so that our experiments are on the same scale as MT-Bench. The question design and types also follow the design of MT-Bench dataset and has a similar distribution. We have included the question types and demos on page 18.
>
> 4. **“Time-stamp cutoffs” for unseen data?**
>
> Thank you for the suggestion. It would be a good component to include in the future when we are evaluating information-rich questions. As most of the current tasks we evaluate (math, coding, writing, roleplay, etc) are examining the model capabilities and not very reliant on external knowledge and new information, we haven’t included such mechanisms so far. But we would be happy to explore it in the future.
>
>
> Thank you again for the helpful comments. Please let us know if the above has clarified some of your concerns or if there are still lingering concerns. We would be happy to address them.

---

> > ### Author Response · Authors · 2024-11-25
> > **Look forward to your reply on our rebuttal**
> >
> > Dear Reviewer MUxT,
> >
> > I hope this message finds you well. We are grateful for your valuable feedback on our submission and are pleased to see your positive score. In our responses, we have addressed the points you raised in detail.
> >
> > As the discussion period is coming to a close soon, we kindly ask if you could review our responses at your earliest convenience. We are eager to know if our explanations have alleviated your concerns. If there are still areas needing improvement, your insights would be greatly appreciated and instrumental in enhancing our work.
> >
> > Thank you once again for your thoughtful review and support.
> >
> > Warm regards, Authors

---

> > ### Author Response · Authors · 2024-11-26
> > **We have revised our paper according to your suggestions**
> >
> > Dear reviewer MUxT,
> >
> > Corresponding to your constructive comments earlier, we have now added the limitations section in L535-539 in the newly uploaded rebuttal revision. Please let us know whether we have addressed your concerns to your satisfaction. Thank you again for your time and efforts.
> >
> > Best regards,
> >
> > Authors

---

> > > ### Comment · Reviewer_MUxT · 2024-11-26
> > > **I would like to keep the score**
> > >
> > > Thank you for the considerations, I would like to keep my score of 6. Thank you.

---

### Official Review · Reviewer_y6Px · 2024-11-03

**Soundness:** 2
**Presentation:** 3
**Contribution:** 2
**Rating:** 6
**Confidence:** 3

**Summary:**

This paper proposes a novel framework for automatically evaluating LLMs using LLM-powered agents. The framework consists of three main components: question generation, multi-round peer battles, and committee discussions. The authors validate their approach through extensive experiments with 15 recent LLMs, achieving a 92.14% correlation with human preferences without manual effort. The paper also reveals some of the intriguing LLM behaviors in competitive peer battles.

**Strengths:**

1. The motivation of the paper is good and clear: current LLM evaluation methods face major limitations. Static benchmarks lack flexibility and may suffer from data contamination, human evaluation requires extensive manual effort, and single-judge evaluations risk bias. This highlights the need for an automated, reliable evaluation method that aligns well with human preferences.

2. The proposed method is validated by comprehensive experiments on 15 modern LLMs, achieves a 92.14% correlation with human preferences, and demonstrates extensibility across languages and domains.

3. The paper is well-structured, well-written, and clearly explained; it is a pleasure to read.

**Weaknesses:**

The application of multi-agent debate in automatic evaluation is not particularly novel, and this paper lacks discussion of a closely related work, ChatEval [1].

There are no formal guarantees on the reliability of the evaluation.

I cannot identify further weaknesses in the paper, but there are several questions I would like the authors to address to enhance the work:

(1) How sensitive are the results to the choice and number of initial committee members and their ranking initialization? Would including low-capability LLMs in the judge committees reduce overall performance?

(2) Can you provide examples of cases where Auto-Arena's evaluation significantly diverged from human preferences? What insights did these cases provide?

(3) How do you ensure that the questions generated by LLMs are sufficiently challenging to evaluate a variety of LLMs as they continue to evolve?

[1] ChatEval: Towards Better LLM-based Evaluators through Multi-Agent Debate

**Questions:**

See the weakness section above.

---

> ### Author Response · Authors · 2024-11-23
> **Thank you for your comments. Please see our responses.**
>
> We thank the reviewer for his/her constructive comments and reply to some of the concerns below:
>
> 1. **“Multi-agent debate in automatic evaluation is not particularly novel”. Discussion of ChatEval.**
>
> Yes, multi-agent debate has been used before for LLM-as-a-judge, which corresponds to the committee discussion component in our framework. We are aware of prior works and discuss them in L513-519.
>
> The main novelty of our paper is the first completely automated evaluation framework, from question drafting to LLM reviewers. Therefore, it innovatively eliminates the need for any manual efforts in the process. Although prior works show that using multi-agent-debate for evaluations could mitigate bias and improve performance, they are still constrained to evaluating LLM performances on existing datasets (manually constructed).
>
> Thank you the suggestion on including ChatEval. In fact, it was included in an earlier version but was deleted later due to the page limit. Currently, we mention DRPE and PRD in L514-517, which are similar to ChatEval and use multi-agent-debate for LLM jugdments. We will include ChatEval in L517 as well!
>
> 2. **“No formal guarantees on the reliability of the evaluation.”**
>
> We agree that, because the method is inherently empirical, it’s not feasible to derive formal guarantees. The extensive study on an extensive set of 15 models provides empirical evidence that the method is robust and expandable.
>
> 3. **“How sensitive are the results to the choice and number of initial committee members and their ranking initialization? Would including low-capability LLMs in the judge committees reduce overall performance?”**
>
> The committee members are updated after each debate according to best 5 models in the current ranking. Therefore, the choice stablizes through the iterations.
>
> In the table presented in Appendix F, we do note that better-performing LLMs can serve as better judges. However, as mentioned previously, as we include more and more LLMs in the ranking, the judge committee will stablize to a group of high-performing ones.
>
> 4. **“Examples of cases where Auto-Arena's evaluation significantly diverged from human preferences?”**
>
> Thank you for this note. There are papers that have specifically studied how LLM-based judges differ from humans. [1] discovers that LLM judges possess Misinformation Oversight Bias, Gender Bias, Authority Bias, and Beauty Bias. In contrast, humans also have strong Misinformation Bias and Beauty Bias, but very minimal Gender Bias.
> As Auto-Arena is also built with LLM-as-a-judge, the differences with human judgments would be quite similar. Although this is not the main focus of the paper, we agree that such case studies would make for an interesting case study and would aim to incorporate such a study on the website or in a later version of the paper.
>
> 5. **“Ensure that the questions generated by LLMs are sufficiently challenging”?**
>
> In our current prompt to the examiner (shown in Appendix A.1), we ask the examiner to devise “complicated and difficult, requiring in-depth understanding and analysis of the subject”. The prompt design ensures that the questions are challenging and debatable.
>
> Moreover, as the LLMs evolve, we will also update the examiner accordingly to be the best-performing LLM to ensure that it has the highest capability itself. Therefore, the question the best-performing LLM deems to be challenging would be challenging to other LLMs, too.
>
> Last but not least, as the most important design is the peer-debate component, the LLM debaters raise questions to each other during the debate process, which democratizes the question generation process. The initial question, in this case, will only serve as seed question/topics for the debate. In the process, LLMs candidates also try to generate the most challenging question it could to further attack its opponents.
>
> Thank you again for the helpful comments. Please let us know if the above has clarified some of your concerns or if there are still lingering concerns. We would be happy to address them.
>
>
> [1] Humans or LLMs as the Judge? A Study on Judgement Biases.[2] Justice or Prejudice? Quantifying Biases in LLM-as-a-Judge.

---

> > ### Comment · Reviewer_y6Px · 2024-11-24
> >
> > I thank the authors for their responses. I would like to maintain my original score.

---

> > > ### Author Response · Authors · 2024-11-26
> > > **Follow-up**
> > >
> > > Dear reviewer y6Px,
> > >
> > > Thank you for the constructive comments again. Just to follow up, we have now added the ChatEval citation (and another similar one, Peer-reviewer-in-LLMs) in the rebuttal revision L509-510.
> > >
> > > Best regards,
> > >
> > > Authors

---

### Author Response · Authors · 2024-11-23
**Please let us know if there are lingering concerns**

Dear reviewers,

We have addressed the review comments in the rebuttal. Once again, thank you so much for the valuable inputs. If there are lingering concerns, do you mind raising them in the discussion? We'd be happy to discuss them further. We look forward to your replies.

Best regards,
Authors

---

### Author Response · Authors · 2024-12-03
**General Conclusion**

We'd like to sincerely thank all the reviewers for their thoughtful reviews and instructive feedbacks. Here, we summarize some of the strengths noted by the reviewers, changes we have made in the rebuttal revision, and our key takeaways once again.

> Strengths
> 1. **Motivation**: "good and clear" (y6Px); important to the field of LLM evaluation (MUxT, y6Px).
> 2. **Method**: "Novel" (MUxT), "Versatile across different languages and tasks" (j7qN, y6Px); "Cheaper" costs compared to "comparable approaches" (PycX).
> 3. **Experiments**: "Solid experimental validation" (PycX); "Comprehensive" (y6Px); "Results well above existing baselines" (MUxT).
> 4. **Paper writing**: "well-structured, well-written, and clearly explained" (y6Px, MUxT)

Corresponding to the feedbacks provided by reviewers, we have made the following changes in the rebuttal revision. No further concerns have been raised.

> Rebuttal Revisions
> 1. For a more in-depth analysis on the evaluation results, we have included a radar chart of model performances in 4 representative categories (writing, reasoning, math, and coding) along with a detailed analysis in Appendix F
> 2. For computational costs, besides the current overall analysis on costs, we have included a detailed breakdown of computational resources in Appendix I. It now includes a table of detailed breakdown costs for each framework component in Auto-Arena and an analysis on the costs.
> 3. To improve the paper, we added references to two more papers: ChatEval and Peer-reviewer-in-LLMs in Related Work L509-510. We have also added discussion on potential limitations in L535-539.

To summarize, our paper has made the following contributions:

> Key Takeaways
> 1. We proposed Auto-Arena, which tackles the LLM evaluation task with full automation in an end-to-end manner. It provides a fast, cheap, and scalable way to evaluate LLMs in the fast-paced industry today.
> 2. As the method is fully automated, it can be easily scaled to other domains and languages. This is especially helpful for usage scenarios where evaluation datasets are scarce.
> 3. We conduct comprehensive experiments and ablations to show the robustness of our method, which demonstrates the highest alignment with human preferences despite zero manual efforts and cheaper costs.

---

### Meta-Review · Area_Chair_hz9E · 2024-12-21

**Metareview:**

This paper introduces Auto-Arena, an automated evaluation framework for large language models (LLMs) that utilizes a multi-stage process comprising question generation, peer battles, and committee discussions. By emulating human-like evaluation, Auto-Arena overcomes the limitations of static datasets and manual assessments. The framework demonstrates a strong correlation with human preferences, improving the reliability of LLM evaluation. Furthermore, Auto-Arena is highly adaptable to various languages and domains, making it a versatile tool for evaluating diverse LLMs and facilitating scalable leaderboard creation.

Positive points:
+ The motivation of the paper is interesting and seems good.
+ The paper is well written

Negative points:
-  The selected topic can be not particularly novel
-  The alignment between LLM and real humans should be further studied
-  The baselines in the experiments can be more comprehensive

**Additional Comments On Reviewer Discussion:**

In the rebuttal period, the authors have provided detailed responses. The reviewers provided feedback on the authors' rebuttal. However, only one reviewer increased the rating to 6. The final ratings of the paper are 5,6,6,6, where there the attitudes are not very consistent, and there is no high score to support the paper. In addition, reviewer  j7qN mentioned that the code is not fully anonymized. At last, I tend to reject the paper.

---

### Decision · Program_Chairs · 2025-01-22

Reject